# Middle East Medicinal Plants in the Treatment of Diabetes: A Review

**DOI:** 10.3390/molecules26030742

**Published:** 2021-01-31

**Authors:** Alaa M. Abu-Odeh, Wamidh H. Talib

**Affiliations:** 1Department of pharmaceutical sciences, Faculty of Pharmacy, The University of Jordan, Amman 11942, Jordan; ala9160555@ju.edu.jo; 2Department of Clinical Pharmacy and Therapeutics, Applied Science Private University, Amman 11931-166, Jordan

**Keywords:** antidiabetic plants, natural products, hyperglycemia, plant extracts

## Abstract

Diabetes is a global health problem, and the number of diabetic patients is in continuous rise. Conventional antidiabetic therapies are associated with high costs and limited efficiency. The use of traditional medicine and plant extracts to treat diabetes is gaining high popularity in many countries. Countries in the Middle East region have a long history of using herbal medicine to treat different diseases, including diabetes. In this review, we compiled and summarized all the in vivo and in vitro studies conducted for plants with potential antidiabetic activity in the Middle East region. Plants of the Asteraceae and Lamiaceae families are the most investigated. It is hoped that this review will contribute scientifically to evidence the ethnobotanical use of medicinal plants as antidiabetic agents. Work has to be done to define tagetes, mechanism of action and the compound responsible for activity. In addition, safety and pharmacokinetic parameters should be investigated.

## 1. Introduction

Diabetes is a major endocrine health problem that has a fast-developing rate around the world. Diabetic patients are anticipated to increase in numbers to 300 million by 2025 according to the World Health Organization (WHO). The Middle Eastern and North African regions have the second highest rates of increase in diabetes globally, with the number of people with diabetes projected to increase by 96.2% in 2035, increasing the social and economic burden of many countries [1,2,3].

The syndrome of diabetes describes a metabolic disorder with disturbances of carbohydrate, fat and protein metabolism, resulting from defects in insulin secretion, insulin action, or both, eventually leading to chronic hyperglycemia and a wide range of complications, including damage to the nervous system, kidneys, blood vessels, eyes, heart, feet and skin [2,4]. For centuries, traditional medicine from plants has been used to treat vast numbers of ailments, including diabetes, as they are considered available and safe [2].

It was documented that 656 flowering plant species are used traditionally for diabetes. Different plant parts were used, including flowers, fruits, seeds, leaves, berries, bark and roots. Plants are selected depending on affordability, stage of progression, comorbidities, availability and safety [5] (https://stateoftheworldsplants.org/2017/).

Recently, several in vitro and in vivo studies have been conducted to study numerous herbs that were claimed to reduce blood glucose level, but of an estimated 250,000 plants, less than 2500 have been studied for pharmacological efficacy against diabetes [5,6].

At present, metformin, obtained from *Galega officinalis*, and the oligosaccharide acarbose, produced by the fermentation of *Actinoplanes utahensi*, are antidiabetic drugs derived from natural origins [6].

Limited efficacy, narrow tolerability, increased side effects and complications, high cost and decreased adherence are the major drawbacks of conventional antidiabetic therapies that raise the necessity to discover new antidiabetic plants [6].

In the present review, an attempt has been made to compile and summarize the reported antidiabetic medicinal plants of the Middle East area. The review also covers the common name of the plant, the parts used, extract type, phytochemical constituents, the test model and suspected mechanism of action, and provides recommendations for future research.

## 2. Medicinal Plants with Potential Antidiabetic Activity in the Arabian Peninsula

The Arabian Peninsula is formulated from the Kingdom of Saudi Arabia, Kuwait, Bahrain, Yemen, Qatar, and the United Arab Emarat, and is located in the Asian southwest [7].

The Kingdom of Saudi Arabia occupies around four-fifths of the Arabian Peninsula, with a population of more than 33.3 million people. The prevalence of diabetes mellitus registered a 10-fold upsurge in the past three decades. In fact, it affects over 25% of the adult population [8]. Oman is ranked 8th in the top 10 countries of the Middle Eastern and North African (MENA) region for diabetes prevalence in 2010 (13.4%) and 2030 (14.9%) [9], while the prevalence of diabetes mellitus is 16.7% in the adult Qatari population [10].

In developing countries, the use of medicinal plants species goes back thousands of years, and forms an important part of the culture, as a large segment of the population still relies on it to treat serious diseases, including diabetes [11].

### 2.1. Oman

#### 2.1.1. *Ajuga iva* (Lamiaceae)

The hypoglycemic effect of the aqueous extract of the whole plant of *Ajuga* was examined in normal and streptozocin diabetic rats. A single oral administration of 10 mg/kg of the extract significantly decreased plasma glucose level over 2–6 h either in normoglycaemic rats or in hyperglycemic rats. However, daily treatment with the same extract for 21 days decreased the plasma glucose level slightly only in the third week in normoglycaemic rats, while diabetic rats showed a significant decrease after the first week and continuously decreased thereafter, supporting its traditional use for diabetes mellitus treatment [12].

On the contrary, the continuous intravenous infusion of the aqueous extract of the whole plant caused significant hypoglycemia in diabetic rats only [13].

The observed pharmacological activity is thought to be related to flavonoids that can enhance insulin secretion, prevent *β*-cell apoptosis, and modulate proliferation. In particular, naringin and apigenin were able to lower blood glucose level significantly, thus showing great potential as antidiabetic agents [14].

Phytoecdysteroids were extracted from *Ajuga iva* and tested for their potential antidiabetic effect in alloxan diabetic rats. The glucose and insulin levels reduced significantly following the administration of phytoecdysteroids. Besides this, they attenuated the metabolic changes caused by diabetes [15]. Their effects may be explained by the stimulation of surviving *β*-cells of islets of Langerhans to release more insulin, controlling the *β*-cell potential [16].

#### 2.1.2. *Moringa pergrina* (Moringaceae)

Previously, *M. peregrina* was reported to have hypoglycemic properties. The antidiabetic activity was reported for the hydroalcoholic extract fraction of *M. peregrina* seeds and aerial parts in streptozocin diabetic rats [17]. Both extracts significantly decreased blood glucose levels comparably to the oral antidiabetic reference drug gliclazide. On the other hand, the *n*-hexane fraction was the only one that showed a highly significant antihyperglycemic activity that was attributed to the lupeol acetate and *β*-sitosterol effect [18].

The ethanol and aqueous extracts of *M. peregrina* seeds at a dose of 150 mg/kg lowered the blood glucose level in diabetic rats and increased the activity of the enzymatic antioxidants, such as catalase (CAT), superoxide dismutase (SOD), glutathione peroxidase and glutathione-*S*-transferase [19].

The hydroalcoholic extract of *M. peregrina* leaves showed inhibitory potential against three in vitro model enzyme assays: *α*-glucosidase, *α*-amylase, and dipeptidyl peptidase IV (DPP IV). The results for pancreatic *α*-amylase suggested that the enzyme responded to the extract when the concentration was increased, moderated intestinal *α*-glucosidase inhibitory potential, and caused a gradual inhibition of the activity of mammalian DPP IV enzyme [20].

The chloroform extract of *M.peregrina* leaves caused a significant decrease in the blood glucose levels in treated mice, probably by increasing peripheral utilization of glucose [21]. 

#### 2.1.3. *Rhazya stricta* (Apocynaceae) 

Recently, it was found that the oral administration of the leaf extract 0.5, 2, and 4 g/kg decreased the plasma glucose level and enhanced insulin level in streptozocin diabetic rats [22,23].

In another study, the effect of different doses of *Rhazya* aqueous extract on adiponectin protein and insulin resistance was analyzed. The data indicated a significant inverse correlation between adiponectin levels and insulin resistance, and a significant increase in adiponectin levels that is considered a promising therapeutic strategy in treating diabetes [23]. 

Crude methanolic leaf extract of *R. stricta* had the best antidiabetic effect compared to other methanolic extracts of different plant parts that were tested in vivo in both male and female albino mice for the reduction of blood glucose and other blood parameters [24]. The leaves extract was fractionated using *n*-hexane, ethyl acetate, chloroform, and water. All fractions were tested for the same activities. The ethyl acetate fraction was the most effective in fasting and random blood as regards the reduction in glucose level, and was comparable to metformin [24].

Later, three Omani traditional medicinal plants, i.e., *Ajuga iva*, *Pteropyrum scoparium*, and *Rhazya stricta*, were tested for their effect on reducing diabetic incidence. The results of the study revealed that the selected plants had reduced the blood sugar level of the treated mice; the extracts of *R. stricta* and *A. iva* had more pronounced effects than *P. scoparium*. The hypoglycemic effect of *R. stricta* extract may be due to its potentiating effect on the insulin-releasing mechanism; this suggests a mode of action resembling the mechanism of sulfonylureas [25].

### 2.2. Qatar

#### *Cynomorium coccineum* (Cynomoriaceae)

Pharmacological studies showed that the *Cynomorium* plants had many biological activities, including antioxidant, immunity-improving, antidiabetic, neuroprotective, and other bioactivities, some of which were already reported by traditional medicine [26].

*α*-glucosidase and *α*-amylase are the key enzymes for controlling the postprandial blood sugar level and managing hyperglycemia. The aqueous extract of *C. coccineum* demonstrated a relatively high *α*-glucosidase inhibitory activity and a moderate inhibition of *α*-amylase, indicating that the edible plant can be a diet-based solution for managing the early stages of diabetes when coupled with other pharmacological management strategies [27].

### 2.3. Saudi Arabia 

#### 2.3.1. *Avicennia marina* (Avicenniaceae)

The ethanolic leaf extract of *A. marina* at doses 250 and 500 mg/kg reduced blood glucose level significantly. It also reduced the level of serum urea that confirms the capacity to protect vital tissues, for example, kidney, and it also improved the biochemical parameters such as serum phosphorous, albumin, and globulin [28]. 

The methanolic extraction of the aerial roots of *A. marina* resulted in the isolation of stigmasterol-3-O-*β*-D-glucopyranoside, which may be responsible for the antihyperglycemic effect seen [29]

The oral supplementation of the ethanolic extracts from *A. marina* leaves 2 mg/gm exerted a significant hypoglycemic effect on streptozocin diabetic mice, similar to the effect of the aqueous and hydroalcoholic extract [30,31,32]. The possible mechanism underlying the antihyperglycaemic action of *A. marina* was attributed to inducing *β*-cells to release more insulin, and reducing oxidative stress by increasing the antioxidant activity of catalase, glutathione-S-transferase, and superoxide dismutase enzymes [28,30,31]. 

#### 2.3.2. *Caralluma sinaica* (Asclepiadaceae) 

The ethanolic extract of *C. sinaica* was evaluated in streptozocin diabetic rabbits. The blood glucose-lowering effect of alcoholic extract 100 mg/kg was more pronounced in diabetic animals than in the glibenclamide-treated group. In addition, it had the capacity to prevent an increase in hyperglycemia after oral glucose load [33]. As the dose increased from 150 to 200 mg/kg, there were no toxic or behavioral changes observed in the animals [33].

The plant extract could reverse weight loss and increase liver glycogen in diabetic rats. This could be consistent with the ability of the extract to restore an adequate insulin level, which prevents the metabolic changes associated with diabetes, lipogenesis and glycogenesis, and the breakdown of muscle proteins [33].

#### 2.3.3. *Ducrosia anethifolia* (Apiaceae)

The ethanolic extract of *D. anethifolia* and its major isolated furanocoumarins demonstrated in vitro inhibitory effects against carbohydrate metabolizing enzymes, *α*-amylase, *α*-glucosidase, and *β*-galactosidase in a concentration-dependent manner [34]. The most potent inhibitors were imperatorin and 5-methoxypsoralen, while psoralen, oxypeucedanin hydrate, and isooxypeucedanin were moderated inhibitors [34]. The biological activity of the *D. anethifolia* ethanol extract showed a hypoglycemic effect that may be related to potentiating the pancreatic secretion of insulin from islet *β*-cells, or the transport of blood glucose to the peripheral tissue, or antioxidant activity in several areas [34].

The antihyperglycemic effect of different isolated furanocoumarin compounds was explained by their stimulatory action as regards glucose uptake by cells and the reduction of oxidative damage of the pancreas [34].

#### 2.3.4. *Jatropha curcas* (Euphorbiaceae)

The chloroform extract of *J. curcus* leaves was reported at doses of 250 and 500 mg/kg for its antidiabetic potential in alloxan diabetic rats; it also caused a reversal in cholesterol, triglyceride, HDL, and LDL values when compared to untreated diabetic rats [35]. 

The hypoglycemic potentials of orally administered aqueous root extract of *J. curcas* 250 and 450 mg/kg were investigated in alloxan diabetic rats [36]. The extract produced a sustained significant reduction in blood glucose level, and elicited anti-infective activity and an ameliorative effect on alloxan-induced anemia [36].

The antidiabetic actions of different extracts of *J. curcas* leaves, petroleum ether, ethyl acetate, and methanol were evaluated in streptozocin diabetic rats. All caused significant improvements in the levels of glucose and *α*-amylase. The histopathological investigation revealed the regenerative and protective effect of extracts on *β*-cells and liver [37].

The antioxidant, antihyperglycemic and ameliorative properties of plant extracts may offer a potential therapeutic source for the treatment of diabetes attributed to the presence of flavonoids [37].

#### 2.3.5. *Loranthus acaciae* (Loranthaceae)

The antidiabetic activity of the crude ethanolic extract and its *n*-hexane, chloroform, and *n*-butanol fractions was investigated in alloxan diabetic rats, and we also performed glucose-tolerance tests in normal rats [38].

The crude extract and the chloroform fraction at a dose of 500 mg/kg had the highest hypoglycemic effect in diabetic rats, with 33.6 and 47% reductions in blood sugar levels. This effect was statistically significant in comparison to the glibenclamide-treated group [38].

*L.acaciae* was thought to compact diabetes through its flavonoids, which are suggested to demonstrate powerful antioxidant activity, and act as inhibitors of biological targets, mostly enzymes such as *α*-glycosidase, *α*-amylase, and dipeptidyl peptidase IV (DPP-4). These results further support the claim for and use of *L. acaciae* in folklore medicine in Saudi Arabia as an antidiabetic drug [38].

#### 2.3.6. *Lyceum shawii* (Solonaceae)

Experimentally, the traditional antidiabetic claim was assessed in normal and streptozocin diabetic rats by using the ethanolic extract of *L. shawii*’s aerial parts. There was a significant hypoglycemic potential in normal rats, as well as hyperglycemic rats that was comparable to the hypoglycemic drug glibenclamide, after both oral as well as intraperitoneal administration [39].

Phytochemical screening helps in explaining the antidiabetic activity of *L. shawii*. Diterpenoids were reported to inhibit *α*-glycosidase. Glycosides, polysaccharides, and saponins were reported to protect pancreatic islets and *β*-cells, and flavonoids were reported to possess glucosidase inhibitory effects and antioxidant activities [39].

The study also investigated the acute and chronic toxic potential of *L. shawii* treatment using mice as an experimental model. There were no alarming signs of acute toxicity, but the plant possessed a significant spermatotoxic potential with chronic use [39].

#### 2.3.7. *Marrubium vulgare* (Lamiaceae) 

After the administration of the *M.vulgare*’s aerial parts’ extract, the elevated plasma glucose levels in diabetic rats were lowered, similar to the effect of glibenclamide. The possible mechanism of the extract was attributed to phenolic compounds, which served as antioxidants, and was achieved through a stimulation of insulin release from the remnant pancreatic *β*-cells [40].

#### 2.3.8. *Moringa oleifera* (Moringaceae)

It was reported that the administration of an aqueous extract of *M. oleifera* manifested potent antihyperglycemic and antihyperlipidemic effects in both insulin-resistant and insulin-deficient rat models [41]. 

The administration of *Moringa* seeds powder to streptozocin diabetic rats caused a drop in fasting blood glucose and increased the antioxidant activity of blood enzymes. However, the antidiabetic activity of the higher dose of the seeds powder (100 mg/kg) was more efficient than that of the lower dose (50 mg/kg) [42].

*M. oleifera* contains three classes of phytochemicals: glucomoringin, phenols, and flavonoids. All of these may contribute to antidiabetic activity through the stimulation of insulin secretion, and the protection of the intact functional *β*-cells from further deterioration or the generation of destroyed *β*-cells [41,42].

#### 2.3.9. *Morus nigra* (Moraceae)

The leaves extract of *M.nigra* decreased glucose and increased insulin levels significantly in diabetic animals [43].

In alloxan diabetic rats, treatment with either *M. alba* or *M. nigra* fruit extracts decreased the hyperglycemia significantly, almost to the normal level. The same effect was reported earlier in streptozocin diabetic mice [44].

The antidiabetic effect can be explained by the inhibition of *α*-glucosidase, *α*-mannosidase, and *β*-galactosidase. N-containing pseudo-sugars were thought to be responsible for this activity. In addition, fagomine strengthened the glucose-induced insulin secretion similarly to the action of the sulfonylurea drug, and increased the tissue uptake of glucose [44].

#### 2.3.10. *Ocimum forskolei* (Lamiaceae)

The *O.forskolei* extract from leaves and stem showed promising results in the *α*-amylase inhibition assay. They almost had the same potential for *α*-amylase inhibition (72.3 and 78.9 µg/mL, for leaves and corollas extracts, respectively), suggesting that *O. forskolei* might be effective in slowing down the hydrolysis of polysaccharides [45]. 

#### 2.3.11. *Plicosepalus curviflorus* (Loranthaceae)

Traditionally, the stem of *P. curviflorus* has been used for the treatment of cancer in Yemen, and the treatment of diabetes in Saudi Arabia [46]. The petroleum ether, ethyl acetate and methanol soluble fractions of *P. curviflorus* were subjected to column chromatography. Two new flavane gallates were found to show significant hypoglycemic activities when tested in rats [46].

The antihyperglycemic activity of *P. curviflorus* and its nanoparticle suspension formulation was evaluated in high-fat diet/streptozocin diabetic rats. The total extracts of *P. curviflorus*, as well as the solid lipid nanoparticle (SLN) formulations, exhibited a significant lowering in blood glucose and insulin resistance, associated at least partly with antioxidant effects, in the diabetic rats and their SLN formulations [47].

#### 2.3.12. *Retama raetam* (Fabaceae)

The administration of methanolic extract of *R. raetam* at 250 or 500 mg/kg daily for 4 weeks showed an appreciable antihyperglycemic effect in diabetic rats, explained based on the inhibition of the renal reabsorption of glucose, as evidenced by increased glycosuria [48].

The plant contains many quinolizidine alkaloids that have been reported to exhibit hypoglycemic activity due to their insulin-releasing properties and antioxidant activity, in addition to the inhibition of glucose absorption that was dose-related [48].

#### 2.3.13. *Rhizophora mucronata* (Rhizosphoraceae)

Both aqueous and hydroalcoholic bark extracts of *R. mucronata* possessed significant hypoglycemic and antihyperglycemic activities attributed to their *α*-glucosidase inhibition potential [28].

The antidiabetic activity of the leaves extract of *R. mucronata* showed promising results in streptozocin diabetic rats. The extract reduced oxidative stress and increased antioxidants activity, and it had an insulin-mimetic effect. Also, flavonoids could play an important role in the prevention of β-cell apoptosis, and the promotion of β-cell propagation, beside the secretion and enhancement of insulin activity [31].

#### 2.3.14. *Salvadora persica* (Salvadoraceae)

The administration of *S.persica* extract decreased blood glucose significantly in the first week of treatment, and this was more evident in the third week for both doses used [49]. *Salvadora persica* contains *β*-sitosterol with reported antioxidant effects, and many amides have been reported as having an insulin secretagogue effect and *α*-glucosidase-inhibition activity [49].

### 2.4. Yemen

Yemen is a small country that occupies an important location in the southwestern part of the Arabian Peninsula. The Yemeni highlands experience relatively high rainfall and have a temperate climate, giving them a high topographic diversity, which along with climatic factors has resulted in a diverse and rich flora [50].

#### 2.4.1. *Azadirachta indica* (Meliaceae)

The water extract of neem leaves was tested in alloxan diabetic rabbits via administration daily for 25 days. The reduction in blood glucose level was significant with regard to control, in a dose- and time-dependent manner [51].

The hypoglycemic effect of neem root bark extract was tested in alloxan diabetic rats, and the reduction in glucose level was significant at high doses only [52].

The oily extract of neem had the potential to reduce the blood glucose levels within a short period time, and it also improved the glucose tolerance after a treatment period of 4 weeks, as suggested by oral glucose tolerance tests in alloxan diabetic rats [53].

Neem’s various parts have been actively used for the treatment of diabetes. Improving the expression of insulin signaling molecules, elevating insulin output, inhibiting epinephrine’s action on glucose metabolism, improving glucose tranporter-4 protein resulting in increased use of peripheral glucose, inhibiting the production of hepatic glucose, and free radical-scavenging properties were all attributed to the antidiabetic mechanism of Neem [54,55,56].

#### 2.4.2. *Boswellia carterii* (Burseraceae)

Ursane, oleanane, and lupine of Olibanum were identified to be responsible for the observed activity in many cases [57]. The pharmacological testing for the water extract in streptozocin-nicotinamide diabetic rats showed a hypoglycemic effect resembling that of glibenclamide and metformin. The effect is possibly due to the stimulation of insulin secretion from the remaining *β*-cells, the antioxidant activity, and the increase in serum insulin [58]. It is thought that *B. carterii* exhibited antidiabetic action through liver glycogen, the inhibition of degenerative changes in the *β*-cells of the pancreas in an alloxan diabetic model, and the suppression of apoptosis of peri-insular cells, and these effects were attributed to *β*-Boswellic acid derivatives [59,60].

#### 2.4.3. *Cissus rotundifolia* (Vitaceae)

The blood glucose data clearly indicated that the aqueous extract from *C. rotundifolia* produced significant hypoglycemic effects in streptozocin diabetic rats. The continuous treatment with 100 mg/kg of *C. rotundifolia* for a period of 28 days produced a significant decrease in the blood glucose levels of diabetic rats [58].

The antidiabetic activity of the plant was tested on healthy human subjects. The healthy volunteers received, in a random order, the control stew meal/control wheat bread and the test stew meals/wheat bread containing cissus flours. Compared with the controls, cissus meals elicited significant reductions in plasma glucose and insulin levels at various postprandial time points, possibly due to starch and water-soluble non-starch polysaccharides [61].

The plant was reported to have quercetin, which is known for its ability to inhibit the insulin-dependent activation of PI3K (Phosphoinositide 3-kinase) and to reduce intestinal glucose absorption by inhibiting glucose transporter [17]. Other species of the same genus, including *C. verticillata* and *C. quadragualis*, possess hypoglycemic and hypolipidemic activities [58,62].

#### 2.4.4. *Dracaena cinnabari* (Dracaenaceae)

The resin of *cinnabari* trunk ethanol extract was tested against the MCF-7 cell line using different solvents for its antidiabetic activity, and the glucose uptake-inducing activity of ethylacetate extract was found to be higher than that of Metformin, which signifies the potential of this extract to be used as a source of antidiabetic drugs [63].

Furthermore, Al-Baoqai studied the antidiabetic potential of the ethanolic extract of *cinnabari* resin for alloxan diabetic rats. Both extract doses of 100 and 300 mg/kg resulted in a significant decrease in fasting blood glucose level, with a recovery in the destruction of pancreas cell [64].

The hypoglycemic activity can be related to flavonoids—the main chemical constituents of the *Dracaena* species—through the inhibition of *α*-glucosidase activity, the suppressing of intestinal carbohydrate absorption and the inducing of glucose uptake activity [64].

#### 2.4.5. *Opuntia ficus-indica* (Cactaceae)

An earlier study compared the effects of an aqueous extract and stem/fruit skin blend prepared from *O. indica* on normoglycemic male Wistar rats. The blood glucose-lowering effect and plasma insulin-increasing effect of the aqueous extract was obvious and statistically significant, but less so than the effect of glyburide. After 15 and 30 min, the optimum effects of increasing plasma insulin and lowering blood glucose were observed, respectively, from a dose of 5.88 mg/kg. On the other hand, the proprietary blend did not affect basal glucose levels in a dose of 6 mg/kg over a period of 180 min, but exhibited a stimulation effect on basal plasma insulin secretion, pointing to its direct action on *β*-cells [65].

This result is supported by Al-Naqeb, who studied the effect of the oil seed in Streptozocin diabetic rats and its molecular mechanisms. Oil extracts exhibited strong antioxidant actives and caused a significant reduction in plasma glucose level in a dose-dependent manner. In addition, the extract elicited an increase in the expression level of the glucose transporter 2 (Slc2a2) gene that is present in the liver, the activity of which would be essential for both glucose secretion and for keeping the intracellular Glu-6-phosphate concentration low, avoiding the permanent activation of glycolytic and lipogenic genes [66].

#### 2.4.6. *Pulicaria inuloides* (Asteraceae)

*α*-amylase and *α*-glucosidase inhibitory studies demonstrated that *Pulicaria inuloides* essential oil caused a concentration-dependent reduction in the percentage of inhibition of *α*-amylase and *α*-glucosidase. The highest concentration of *Pulicaria inuloides* (250 μg/mL) showed a maximum inhibition of nearly 85.50% and 86.52% of *α*-amylase and *α*-glucosidase, respectively [67].

To confirm the hypoglycemic effect, diabetic rats were treated with a 400 mg/kg (body weight) dose of essential oil orally for 21 days. The *P. inuloides* treatment given to the diabetic group was able to significantly lower the blood glucose level similarly to the standard glibenclamide [67].

The antihyperglycemic effect may be accounted for via several mechanisms, such as its ability to impair the absorption of glucose in the intestine through *α*-glucosidase inhibition, increase glucose uptake from the bloodstream and oxidation in the peripheral tissues, enhance insulin sensitivity, control lipid metabolism thereby fixing the putative inhibition of insulin signaling, and scavenge the free radicals, resulting in increases in the plasma membrane receptors or transporters necessary for the signaling and uptake of glucose from the bloodstream [67].

*P. inuloids* is a safe and effective intervention for diabetes, can correct the metabolic disturbances associated with diabetes, and can reinforce the healing of the liver according to the histopathological studies [68].

#### 2.4.7. *Solenostemma argel* (Asclepiadaceae)

The blood glucose level of methylprednisolone-treated hyperglycemic rats was determined after daily oral administration of 1 g/kg of Argel extract for 10 days, or with glibenclamide at 6 mg/kg. The oral administration of the extract produced a significant reduction in fasting serum glucose in diabetic rats that was comparable to the glibenclamide group [69].

Serum alanine aminotransferase (ALT) and aspartate aminotransferase (AST) were significantly lower in the Argel group compared to the control group, indicating a possible hepatoprotective effect caused by the extract. On the other hand, the administration of glibenclamide resulted in the significant elevation of serum ALT and AST levels, which indicates its side effects [69].

A similar effect was observed in diabetic albino rats treated with water extract of *S. argel*. The author reported the extract’s ability to affect *α*-amylase activity after 2 h [70] (Table 1, Table 2 and Table 3).

## 3. Medicinal Plants with Potential Antidiabetic Activity in Egypt

The International Diabetes Federation (IDF) listed Egypt among the world’s top 10 countries in the number of patients with diabetes. It is expected this number will jump up to 13.1 million by 2035 [126].

There are abundant and biodiverse medicinal and aromatic plant kinds in Egypt. An Egyptian large-scale bio-study was performed, which aimed to screen 264 plant extracts for their in vitro *α*-glucosidase inhibitory activity. Of all extracts, 63 achieved an inhibition of *α*-glucosidase of more than or equal to 70% at the tested concentration (25 ppm, and the most active plant extract is *Pinusrox burghii* (IC50 is 2.47 ppm) [127,128].

### 3.1. Cassia acutifolia (Fabaceae)

Several allied species had been shown to lower blood glucose level. The hydroethanolic extract of *C. acutifolia* leaves showed a high antihyperglycemic effect at dose levels 10 and 50 mg/kg. An anthraquinone, chrysophanol, was isolated from the leaves of *C. acutifolia* and showed mild antidiabetic properties in cell culture. Its activity was proposed to be mediated through affecting glucose transport and the tyrosine phosphorylation of the insulin receptor, improving insulin action or insulin-independent effects [129].

### 3.2. Centaurea alexanderina (Asteraceae)

A significant decrease in elevated blood glucose level was seen in normoglycemic and streptozocin diabetic rats. The methanol extract at a dose level of 600 and 300 mg/kg showed a significant reduction in plasma glucose level after thirty days of treatment, and the maximum effect was observed after 60 days of treatment [130].

### 3.3. Cyperus laevigatus (Cyperaceae)

The biochemical markers caused a decrease in glucose level and the promotion of serum insulin in the diabetic group treated with *C. laevigatus* extract, and this can be explained by flavonoid activity. In addition, the histological examination of the pancreas of the extract-treated rats indicated the normal architecture that was attributed to flavonoids’, flavonoid glycosides’ and phenolic acids’ abilities to regenerate *β*-cells [131].

### 3.4. Fraxinus ornus (Oleaceae)

The hydroalcoholic extract of *F.ornus* fruit showed a significantly antihyperglycemic effect. Improving insulin action or insulin-independent effects was postulated as a possible mechanism for the antidiabetic effect [129].

### 3.5. Phoneix dactylifera (Arecaceae)

It was reported that the oral administration of date seed extract combined with insulin had an antihyperglycemic effect as compared to seed extract alone in streptozocin diabetic rats, and could minimize the toxic effects of diabetes on the liver and kidney for diabetic rats [132].

Good glycemic control was achieved by the administration of an aqueous suspension of *P. dactylifera* seeds. It resulted in a significant reduction (by 51%) in the blood glucose level compared with the untreated diabetic group. In addition, it is reported to possess a protective effect against diabetic complications both for kidney and liver [133]. The hydroalcoholic extract of *P. dactylifera* leaves had a strong inhibitory effect against *α*-glucosidase, and a significant antidiabetic activity superior to the antidiabetic drug acarbose in alloxan diabetic rats [134].

### 3.6. Nepeta cataria (Lamiaceae)

All crude extracts of *N. cataria*, ethanol, petroleum ether, and chloroform extracts revealed significant amelioration in blood glucose and insulin levels, and an improvement in hepatocytes and pancreas *β*-cells, for diabetic rats. The hypoglycemic action of the extract was explained by the presence of antioxidants such as flavonoids and polyphenols, which may prevent the progressive impairment of pancreatic *β*-cell function, and may improve all carbohydrate brush border hydrolyzing enzymes [135].

### 3.7. Securigera securidaca (Fabaceae)

Previously, the aqueous extract of *S. securidaca* seeds showed a significant decrease in blood glucose level in glucose-loaded mice and alloxan diabetic rats. The hypoglycemic effect of the seed was estimated to be related to its flavonoid content [136].

The ethanolic extract of the flowers showed significant antidiabetic activity with a potency comparable to gliclazide in alloxan-induced diabetic rats [136].

The mechanism of hypoglycemia was attributed to several phenolic compounds; Vicenin-2 was reported to be an antioxidant that strongly inhibited *α*-glucosidase, isoquercetrin and astragalin were found to be glycation inhibitors, rutin was reported to enhance peripheral glucose utilization by skeletal muscle, the stimulation of *β*-cells, quercetin and kaempferol were found to improve insulin-stimulated glucose uptake in mature adipocytes, rutin and hesperidin were reported to increase hepatic glycolysis and glycogen concentration and lower hepatic gluconeogenesis, and catechin showed potential insulin-mimetic activity [136].

### 3.8. Trigonella stellate (Fabaceae)

The ethanol extract of *T. stellate* did not show any significant activation of PPAR*α* and PPAR*γ* (peroxisome proliferator-activated receptor *α*/*γ*), but the ethyl acetate fraction showed activation of both genes. On other hand, the isolated compounds, methoxyisoflavan and dimethoxyisoflavan derivatives, showed an increase in PPAR*α* activity, while the other compound, glucopyranosyl isoflavan derivative, showed an ability to activate the PPAR*γ* receptor. The other reported isolated compounds manifested a mild to moderate activation of PPAR*γ* receptors [137].

### 3.9. Urtica pilulifera (Urticaceae)

The ethyl acetate and chloroform fractions of *U. pilulifera* extract decreased the glucose level significantly in diabetic rats. Previously, seeds lectin of *U. pilulifera* was reported to mimic insulin actions by interacting with the glycoprotein residues of the insulin receptor [138].

Other bioactive compounds may contribute to the antidiabetic activity, for example, *β*-sitosterol, *β*-amyrin and ursolic acid through increasing glucose utilization and metabolism in peripheral tissue [138].

### 3.10. Zizyphus spina-christi (Rhamnaceae)

The butanol extract from *Z. spinachristi* leaves, and its main saponins glycoside and christinin-A, improved the oral glucose tolerance and potentiated glucose-induced insulin release in type II diabetic rats. The sulfonylurea-like activity of the extract was reported to have a safe LD50 of 3820 mg/kg when compared to glibenclamide [139].

In another study, the oral administration of *Z. spina-christi* leaf extract, plain or formulated for 28 days, reduced blood glucose level along with a significant increase in serum insulin level and antioxidant capacity. A normalization of the percentage of glycated hemoglobin (HbA1C %) and a dose-dependent inhibitory activity of the extract against *α*-amylase was also reported. *Z. spina-christi* leaf extract improved the glucose level in diabetic rats by increasing insulin secretion, which may be due to both saponin and polyphenols contents; attenuating meal-derived glucose absorption, which might be attributed to the total polyphenols; and restoring liver and muscle glycogen content, together with significantly decreasing hepatic glucose-6-phosphatase and increasing glucose-6-phosphate dehydrogenase activities [140] (Table 4, Table 5 and Table 6). 

## 4. Medicinal Plants with Potential Antidiabetic Activity in Iran

The burden of diabetes is growing in Iran. The prevalence of type 2 diabetes was 11.4% of the adult population, and it was estimated that the rate would increase to 12.8% (9.2 million) by 2030 [159,160]. Iran has a great diversity of medicinal plants due to the specific climate conditions [161].

It is now believed that *α*-amylase and *α*-glucosidase play an important role in controlling diabetes mellitus, especially in patients with type-2 diabetes. Several plants have been recommended in traditional Iranian medicine to treat diabetes [162].

Various extracts, such as *n*-hexane and ethyl acetate, and the methanol of various parts of *Allium paradoxum*, *Buxus hyrcana*, *Convolvulus persicus*, *Pimpinella affinis*, *Parrotia persica*, *Primula heterochroma*, *Ruscus hyrcanus* and *Smilax excelsa*, were examined for *α*-glucosidase and *α*-amylase inhibition. These plants mostly serve as food flavoring. Some extracts of *S. excels*, *P. persica*, and *P. heterochroma* exhibited significant antidiabetic activities in *α*-amylase and *α*-glucosidase assays, which were even more effective than acarbose. In addition, *C. persicus* and *P. heterochroma* showed strong antioxidant activity, compared with butylated hydroxytoluene [163].

Dichloromethane, *n*-hexane, chloroform, ethyl acetate, and methanol were used to prepare various extracts of the plants; *Cinnamomum zeylanicum*, *Crataegus oxyacantha*, *Hibiscus sabdariffa*, *Morus alba*, *Portulaca oleracea*, *Rubus fruticosus*, *Syzygium aromaticum*, *Teucrium polium*, *Trigonella foenum-graecum* and *Vaccinium arctostaphylos* were tested for their inhibition of *α*-glucosidase, *α*-amylase, and antioxidant activity [163]. *S. aromaticum* methanolic extract exerted the highest inhibitory effect against both *α*-glucosidase and *α*-amylase enzymes. Previously, the aqueous extracts were reported to have insulin-like effects, such as increasing glucose uptake into adipocytes. Among the other analyzed plants, *C. zeylanicum*, *H. sabdariffa*, *R. fruticous*, *C. oxyacantha*, *V. arctostaphylos,* and *M. alba* exhibited strong inhibitory activities against *α*-glucosidase in comparison with the reference drug, acarbose [164].

Other plant extracts were screened for *α*-amylase inhibitory activity, and these were *Juglans regia*, *Olea europaea*, *Camellia sinensis*, *Coriandrum sativum*, *Trigonella foenum-graecum*, *Urtica dioica*, *Urtica pilulifera*, *Arctium lappa*, *Calendula officinalis*, and *Hibiscus gossypifolius* [162].

The *α*-amylase-inhibitory activity varied among the tested plant extracts, but all of them demonstrated a significant dose-dependent reduction effect on the *α*-amylase enzyme. The most potent inhibitions appeared to correlate with the extracts of *Camellia sinensis*, *Trigonella foenum-graecum* and *Urtica dioica* leaves, and of *Trigonella foenum-graecum* seeds [162].

The genus *Salvia* generally comprises a variety of phenolic metabolites, especially flavonoids. Bioassay-guided fractionation of *Salvia virgate* extract led to the isolation and identification of an active flavone compound, chrysoeriol. The compound concentration-dependently inhibited the *α*-amylase activity with an IC_50_ value 1.27 mM [165] (Table 7).

The genus allium contains more than 500 species, of which 93 are known from Iran and only a few of them are used as a foodstuff. The most important species existing in this genus include garlics, onions, and leek, which have long been used as spices and for medicinal purposes, including reducing the risk of cardiovascular disease and diabetes, stimulating the immune system, protecting against infections, and exhibiting anti-aging as well as anti-cancer properties. These biological effects of *Allium* vegetables are mainly associated with organosulphur compounds [179,180,181].

### 4.1. Allium ampeloprasum (Liliaceae)

The essential oils of the green parts of Egyptian *A. ampeloprasum*—which are not edible—were tested for their diabetic activity on streptozocin diabetic rats. The extract showed a significant decrease in glucose level and an improvement in lipid profile and oxidative stress parameters. Further studies are required to identify the active constituents [182].

Recently, alloxan diabetic rats were treated with a hydroalcoholic *A. ampeloprasum* extract that showed a treatment effect for hyperglycemia in rats and significantly decreased cholesterol and triglycerides, which is consistent with previous studies [183].

It is well known that the imbalance between reactive oxygen species production and metabolism ends in a range of disorders, including diabetes. The antioxidant power of *A. ampeloprasum* extract is more marked than that of other plants from this genus due to its significantly larger amounts of polyphenolic compounds, phenolic acids, flavonoids, tannins, and saponin than other plants from the same genus. Therefore, these compounds may play an essential role in maintaining the integrity of pancreatic *β*-cells [183].

Generally, plants from the genus *Allium* can cause a decrease in glucose level in an experimentally induced diabetes model, or after glucose loading through increasing the peripheral consumption of glucose, inhibiting the intestinal absorption of glucose, or intensifying the insulin secretion from residual *β*-cells [183].

### 4.2. Allium ascalonicum (Liliaceae)

Thiosulfinates volatile sulfur compounds are typical of the *Allium* species and are reported to cause many of the biological effects of garlic, and are also responsible for their characteristic pungent aroma and taste [166].

*A.ascalonicum* methanolic extract decreased blood glucose level in alloxan diabetic rats in a way that resembles the hypoglycemic action of glibenclamide. In the long-term period, the effect of the reduction in blood glucose was similar to that of metformin. The *Allium* genus is rich in flavonoids that can inhibit the enzymes responsible for the controlling of gluconeogenesis, glucokinase and glucose 6-phosphatase, and can increase the storage of glucose in the liver with a reduction of glycogen breakdown [166].

### 4.3. Allium sativum (Liliaceae)

Several experiments were performed to test the hypoglycemic activity of the plant. In most of the studies, garlic had been found to be effective in lowering the serum glucose level in streptozocin as well as alloxan-induced diabetic rats, mice, and rabbits [184].

In 1996, Augusti and Sheela showed that S-allyl cysteine sulphoxide and allicin had the potential to reduce the diabetic condition in rats almost to the same extent as did glibenclamide and insulin. In addition, both garlic oil and diallyl trisulphide were reported to improve glycemic control in streptozocin diabetic rats [185].

The aqueous extract for fresh bulbs and seeds of garlic administered orally or by injection showed a significant decrease in serum glucose levels in streptozocin diabetic rats in all the reported studies [184,186,187]

Furthermore, *Allium sativum* methanolic extract decreased blood glucose levels in alloxan diabetic rats similarly to glibenclamide. It reduced blood sugar similar to metformin for a long-term period [166].

The proposed mechanisms for the hypoglycemic effect of garlic were the stimulation of insulin secretion, the enhancement of glucose utilization, the inhibition of the intestinal absorption of glucose, and a sparing insulin effect [186].

### 4.4. Amygdalus lycioides (Rosaceae)

The effect of the hydroalcoholic *A. lycioides* extract was evaluated on diabetic rats. The glucose serum level and glucose tolerance test showed a decrease after treatment with plant extract (1000 mg/kg), and the total number and numerical density of *β*-cells were increased [188].

Flavonoids such as quercetin were reported to inhibit glucose absorption in the intestine, and stimulate the insulin secretion and regenerate the *β*-cells of the pancreas, a hypothesis that was confirmed by the stereological studies and the exhibition of strong antioxidant scavenging activity; all are possible mechanisms of the *Amygdalus lycioides* antidiabetic activity [188].

### 4.5. Amygdalus scoparia (Rosaceae)

The daily oral administration of 200 mg/kg of *A. scoparia* extract for 15 days decreased the blood glucose concentration to the normal range in streptozocin diabetic mice, slightly increased the size of pancreatic islets, and markedly regenerated *β*-cells [189].

### 4.6. Arctium lappa (Asteraceae)

The *A. lappa* root extract had an antidiabetic effect through its hypolipidemic and insulinotropic properties. Flavonoids that are known as bioactive antioxidant and antidiabetic agents, have an alkaloid content that can modulate insulin secretion, and have saponins that have blood glucose-lowering effects, and all are responsible for the reported activity [190].

According to the insulin-related biomarkers assay, it was indicated that the *A. lappa* root extract had improvement effects for type 2 diabetes complications through the enhancement of *β*-cells function, the induction of insulin sensitivity and insulin secretion, and a reduction in the insulin-resistance index [190].

### 4.7. Berberis integerrima (Berberidaceae)

The aqueous extract of *B. integerrima* root was tested on streptozocin diabetic rats. Different doses of the aqueous extract resulted in a significant decrease in blood glucose and lipid profile, while HDL-cholesterol was markedly increased [191]. On the contrary, the daily administration of aqueous fruit extract did not possess a hypoglycemic and hypolipidemic activity in streptozocin diabetic rats [192].

Moreover, the antihyperglycemic effect of the anthocyanin fraction of *B. integerrima* fruits in normal and streptozocin diabetic rats was investigated, and the synergic effect of this fraction with metformin or glibenclamide was evaluated [193]. The blood glucose level was significantly decreased in treated diabetic rats. Nevertheless, there was no synergistic effect [193].

Berberis species are rich in anthocyanins that can protect the pancreatic *β*-cells against oxidative stress through antioxidant properties, promote insulin release from the pancreatic *β*-cells, activate AMPK (5′-adenosine monophosphate-activated protein kinase), the main enzyme for enhancing glucose transport into skeletal muscles, and inhibit *α*-glucosidase in the small intestine; all of these are the recognized mechanisms of anthocyanins antidiabetic activity [193].

### 4.8. Brassica napus (Brassicaceae)

The administration of raw and cooked *Brassica napus* extract to alloxan diabetic rats significantly reduced blood glucose compared to diabetic control rats [194].

*Brassica napus* is rich in anthocyanins, which play a role in decreasing and/or inhibiting *α*-glucosidase and inducing insulin secretion via the stimulating of the pancreas *β*-cells [194]. It also contains sulfur-containing amino acids that have a role in glucose-lowering function, the enhancement of insulin’s effect on the body, and the increase in liver glycogen synthesis in diabetic rats [194].

### 4.9. Brassica rapa (Brassicaceae)

Turnip leaf extract significantly decreased serum glucose and prevented the elevation of plasma ALT in a dose-dependent manner. This activity may be due to the possession of high levels of polyphenolic compounds and the presence of flavonoids and tannins. Therefore, turnip leaf chemical components may have exerted a regenerative effect on *β*-cells and stimulated these cells to produce more insulin, or have contained some insulin-like substances [195].

### 4.10. Capparis spinosa (Capparaceae)

Several experimental studies have confirmed the antidiabetic properties of aqueous and hydroalcoholic extracts of *C. spinosa*. The antihyperglycemic effect was observed for different parts of the plant, including fruit, leaves, root, and seeds, and in a wide range of doses and treatment periods [196,197,198,199].

Both Jalali et al. and Eddouks et al. reported the glucose-lowering effect of the aqueous extract of the *C. spinosa* fruit in streptozocin diabetic rats after the oral administration of 20 mg/kg. The effect was undetectable in normoglycemic rats [200].

The antihyperglycemic effects of *C. spinosa* are due to its inhibition of *α*-amylase activity, the reduction in the mRNA expressions, and thew activities of PEPCK (Phosphoenolpyruvate carboxykinase) and G6Pase (Glucose 6-phosphatase) that cause a reduction in basal endogenous glucose production by the liver, the stimulation of intracellular insulin signaling pathways and the enhancing of glucose uptake in the liver, muscle, and adipose tissue, causing an improvement in insulin sensitivity in these tissues [196].

### 4.11. Centaurea bruguierana (Asteraceae)

The hypoglycemic effect of *C. bruguierana* was demonstrated using various extracts of the fruiting aerial parts [201].

All the extracts decreased blood glucose nearly equally to glibenclamide, and reached a steady state after 3 h. The aqueous extract’s hypoglycemic effect continued to increase significantly after 3 h. The plants worked through increasing hepatic glycogenolysis [201].

### 4.12. Cichorium intybus (Asteraceae)

The whole plant ethanolic extract significantly attenuated the serum glucose level by reducing hepatic glucose-6-phosphatase activity [202]. Ethanolic seed extract, fruit and leaf powder were all reported to improve glycemia in rats [202,203,204].

The aqueous seed extract of *C. intybus’* effects on blood sugar and some blood parameters were investigated for detailed differences between early and late stages of diabetes in the diabetic male rats. The treatment with chicory extract over four weeks prevented weight loss in both early-stage and late-stage diabetic rats, and the levels of cholesterol, triglycerides and HbA1c were decreased [203].

The main observation in Chicory-treated diabetic animals was the resistance to excessive increases in fasting blood sugar. In addition, chicory treatment led to an increase in insulin level in the early stage of diabetes, pointing toward the insulin-sensitizing action of chicory [203].

Chicory may be useful as a natural dietary supplement for slowing down the pace of diabetes’ progress due to caffeic acid and chlorogenic acid presence. Both have the potential for increasing glucose uptake in muscle cells, and stimulating insulin secretion from an insulin-secreting cell line and islets of Langerhans [203].

A new potential antidiabetic agent present in the plant is Chicoric, which was reported to exhibit both insulin-sensitizing and insulin-secreting properties [203].

### 4.13. Citrullus colocynthis (Cucurbitaceae)

Various parts of the plant, such as the root, fruit, rind, and leaf, were used to prepare ethanolic, methanolic, or aqueous extracts at varying doses from 10 to 500 mg/kg, and all elaborated the plant’s antiglycemic activity [205].

Furthermore, the plant is widely used traditionally for the treatment of diabetes in Iran [206]. The extract of the plant seed was able to reduce blood glucose level significantly, and to enhance the regeneration of *β*-cells and increase the size of the pancreatic island in alloxan diabetic rats [207].

The administration of fruit powder significantly decreased lipid parameters and glucose level, and increased inulin level [204].

The antidiabetic effect of *C. colocynthis* extract could be linked to the plant’s ability to stimulate the *β*-cells, to activate the insulin receptors, to decrease gluconeogenesis, to inhibit the release of counter-regulatory hormones, to inhibit the effect of glucose absorption, to increase the incorporation of circulating glucose as hepatic glycogen, and to partially regenerate or preserve the pancreatic *β*-cell mass [204,207].

### 4.14. Cornus mas (Cornaceae)

The promising efficacy of *C. mas* in the modulation of blood glucose, as well as lipid parameters, in alloxan diabetic rats was reported. A plausible mechanism is the inhibition of *α*-glucosidase by the acylated anthocyanins found richly in the fruit. Oleanolic acid is a triterpenoid considered to be part of the observed effect, acting by reducing glucose absorption and enhancing the release of acetylcholine from nerve terminals at muscarinic M3 receptors in the pancreatic cells, and it also augments insulin release [208].

### 4.15. Cucumis sativus (Cucurbitaceae)

The oral administration of ethanol extract from fruit or the methanolic fruit pulp extract of *C. sativus* exhibited significant antidiabetic effects in streptozocin or alloxan diabetic rats [209,210,211].

In Asian traditional medicine, *C. sativus* seed has been used as a suitable functional food for medical purposes, such as in diabetes and hyperlipidemia, and as a diuretic for the treatment of hypertension, gall bladder stones, constipation, and dyspepsia [212].

The saponin- and steroid-rich fractions of butanolic extract and the hydroalcoholic total extract were investigated for their pharmacologic action in normal and diabetic rats. It was found that none of the fractions were able to cause hypoglycemia in normal groups, but both applied extracts were effective in lowering blood glucose in diabetic animals [212]. It was postulated that *C. sativus* seed extracts have biguanides-like or euglycemic effects in diabetic condition [212].

### 4.16. Cucurbita pepo (Cucurbitaceae)

The results obtained indicated that the administration of raw pumpkin fruit powder for four weeks in diabetic rats significantly reduced blood glucose level and had a pancreatic protective effect [213].

### 4.17. Eryngium caucasicum (Apiaceae)

Fresh leaves are used as a food additive, flavoring and cooked vegetable in the preparation of several local foods [214].

In one study the fresh leaves of *E. caucasicum* were investigated in streptozocin-nicotinamide diabetic rats. The result showed that *E. caucasicum* extract significantly decrease fasting blood sugar in high doses (200 and 300 mg/kg), and improved insulin secretion significantly [215].

### 4.18. Eucalyptus globulus (Myrtaceae)

In streptozocin diabetic mice, the incorporation of *E. globulus* aqueous leaf extract in the diet (62.5 gm/kg) and drinking water (2.5 g/L) reduced hyperglycemia in a dose-dependent manner, with the partial restoration of pancreatic *β*-cells. In addition, the hypoglycemic effect was seen in alloxan diabetic rats treated with the plant extract. The antihyperglycemic effect with the improvement in insulin level was reported in streptozocin rats fed with plant leaf powder [129,216,217].

*E. globulus* possessed an antioxidant property. It reduced oxidative stress mostly by reducing the plasma glucose level in diabetic rats, thereby preventing the excessive production of free radicals through glycation of the proteins. It affected the glucose metabolism in fat and muscle cells, and decreased blood sugar by increasing the glucose influx in the cells [217].

### 4.19. Falcaria vulgaris (Apiaceae)

Several doses of 200, 600, and 1800 μg/kg of the plant aqueous extract caused a significant decrease in blood glucose similarly to glibenclamide [218]. The antidiabetic effect was assumed to be due to the plant’s ability to improve the diameter of the *β*-islet of the pancreas and promote insulin secretion [219].

### 4.20. Ferula assafoetida (Apiaceae)

*Asafoetida* extract had an antidiabetic effect both on alloxan and streptozocin diabetic rats. The aqueous extract of the plant significantly reduced blood glucose and increased insulin level in alloxan diabetic rats. The possible mechanisms of its effect were potentiating insulin release at lower doses of the extract, preventing intestinal *α*-glucosidase, and activating glucokinase [220,221,222].

Ferulic acid is the main phytoconstituent in *F. assafoetida*. It can increase the activity of glucokinase, hinder glucosidase inside the gut, and inhibit oxidation response [220,221]. In addition, umbelliferone had an antihyperlipidemic potential. It decreased plasma glucose, accelerated insulin level, and decreased insulin resistance, all of which were better under diabetic conditions [221]. The alternative factor is flavonol quercetin, which may probably control hyperglycemia and growth hexokinase, reduce glucose-6-phosphatase and fructose-bis phosphatase sports, and possesses *α*-glucosidase interest [221].

### 4.21. Galega officinalis (Fabaceae)

The oral administration of *G. officinalis* leaf powder to diabetic rats decreased the blood glucose levels to normoglycaemic values only at the higher dose of 3 mg/kg [223].

*G. officinalis* hydroalcoholic extract was able to greatly control the blood sugar in diabetic rats. The reasons for its antihyperglycemic effects were explained by its increasing of the insulin level and its protection of *β*-cells against oxidative stress due to the presence of alkaloids, flavonoids, glycosides, resin, steroids, tannins, and phenols, which, according to the research conducted, have an antidiabetic trait [224].

### 4.22. Gundelia tournefortii (Asteraceae)

The aqueous root, shoot, and aerial part extracts all demonstrated a significant hypoglycemic effect in diabetic mice. The possible mechanisms of antidiabetic activity include improving the pancreas, increasing the number and average diameter of the islets of Langerhans, increasing the secretion of insulin from the pancreas, increasing the insulin sensitivity, activating the peroxisome proliferator-activated receptor (PPAR) that regulates the transcription of the gene involved in lipid and glucose homeostasis and metabolism within the cell, decreasing the glucose absorption from the intestine through decreasing the processes of glucose transport or the inhibition of *α*-amylase, and increasing antioxidant enzymes [225,226].

### 4.23. Hordeum vulgare (Poaceae)

None of the Barley extracts, hydroalcoholic extracts or protein-enriched fractions were able to cause hypoglycemia in normal rats, even after an extended period of treatment, but were able to reduce blood glucose level in diabetic rats after the subacute phase [227].

A possible mechanism for the antihyperglycemic effect of barely might be the antioxidant capacity of the minerals which are abundant in barley seeds, and the presence of small proteins and essential amino acids and their ability to induce insulin release or amplify its secretion, as induced by glucose [227].

### 4.24. Juglans regia (Juglandaceae)

The consumption of a hydroalcoholic dose-dependent extract of walnut leaves was reported to decrease the level of blood glucose in diabetic rats at a level comparable to metformin [228].

It was reported that the consumption of walnut oil for six weeks significantly reduced several biochemical parameters, including blood glucose, insulin, HbA1C, total cholesterol and triglyceride, and the size of Langerhans islets, in alloxan diabetic rats [229].

A hydroalcoholic extract of walnut leaves administered to diabetic rats also contributed to a rise in insulin levels through improvements in the Langerhans islets and *β*-cell restoration [230].

In another study, the hydroalcoholic extract of walnut leaves was reported to cause changes in streptozocin diabetic rat pancreatic tissue, leading to improved Langerhans islets [231].

The antidiabetic activities of walnuts were associated with the release of insulin from the remaining *β*-cells or replicated *β*-cells, the rise in insulin sensitivity, the interference with the absorption of dietary carbohydrates in the small intestine, and the peripheral tissue usage of glucose [228,232].

### 4.25. Mentha spicata (Lamiaceae)

The leaves aqueous extract of *Mentha spicata* caused a significant reduction in blood glucose level caused by the prevention of free radical formation induced by alloxan. The leaves extract of this plant increased the activity of serum non-enzymatic antioxidants and erythrocyte antioxidant enzymes, and markedly decreased lipid peroxidation in erythrocytes and plasma in diabetic rats [233].

### 4.26. Nasturtium officinale (Brassicaceae)

It has already been shown that the ethyl acetate extract of aerial parts of *N. officinale* decreased the blood glucose level in diabetic rats after two months of treatment. On the other hand, the aqueous and methanolic extracts of aerial parts of *N. officinale* did not affect blood glucose levels after a one-week treatment. The conflicting results could be due to the type of the extract and duration of the treatment [234].

In another study, the hydroalcoholic extract of watercress leaves significantly reduced serum glucose, total cholesterol, and LDL (low density lipoprotein) [234].

A feasible mechanism of the hypoglycemic effect is the stimulation of Langerhans islets, the development of peripheral sensitivity to remnant insulin, and the antioxidant houses of watercress. [234].

### 4.27. Otostegia persica (Lamiaceae)

Reductions in insulin resistance, glucose and triglycerides concentrations, and increases in *β*-cells regeneration after treatment with aqueous extract of *O. persica*, were reported [235].

Several experiments revealed that the methanol and ethanol extracts of *O. persica* possessed antihyperglycemic action against streptozocin diabetic rats. This was mediated through the stimulation of insulin release, the improvement of the pancreas tissue, the stimulation of glucose utilization in peripheral tissues, the modulation of hormones regulating carbohydrate metabolism, the inhibition of the proximal tubular reabsorption mechanism for glucose in the kidney, and the inhibition of the *α*-glucosidase enzyme [235].

Stereological methods were used to investigate the effects of *O. persica* extract on pancreatic *β*-cells in streptozocin diabetic rats. The study found a significant reduction in pancreatic weight and volume, and pancreatic islet volume, in diabetic and extract-treated groups, although these reductions were significantly less significant in treated groups, indicating the ability of the plant to prevent the remaining *β*-cells in the pancreatic islets from undergoing some pathologic changes, such as hypertrophy [236,237].

### 4.28. Pyrus boissieriana (Rosaceae)

Wild pear leaf extract was investigated for antihyperglycaemic, antihyperlipidemic and antioxidant effects in alloxan diabetic rats. Both high and low doses of the *Pyrus* leaf extract significantly limited the increase in serum glucose concentration and enhanced serum insulin levels compared to the control group [238,239].

The methanol extract of *P. boissieriana* leaves and all extracts of its stem (*n*-hexane, ethylacetate, and methanol) inhibited *α*-glucosidase more effectively than acarbose [163].

Another in vitro experiment showed that *P. boissieriana* aqueous methanolic leaf extract was able to inhibit both *α*-amylase and glucosidase significantly. Arbutin, a characteristic of the genus *Pyrus* and a major compound in the *P. boissieriana* leaves, also showed the strong inhibition of *α*-amylase and *α*-glucosidase, though less potently than the extract [240,241].

### 4.29. Phoenix dactylifera (Arecaceae)

Both diosmetin 7-*O*-b-*L*-arabinofuranosyl apiofuranoside and diosmetin apiofuranoside caused a marked improvement in the blood glucose homeostasis in diabetic rats. These compounds work through the regeneration of the endocrine pancreas, increasing insulin secretion and stimulating the enzyme glycogen synthetase [158].

Several studies determined the inhibiting capabilities of the glucose absorption of different plant parts. The parthenocarpic, pit, fruit, seed, and leaf extracts prevented glucose absorption by *α*-amylase and *α*-glucosidase inhibition [134,242,243,244].

The effectiveness of the hydroalcoholic extract of Phoenix dactylifera leaves was evaluated in an animal model of type II diabetes. The postprandial hyperglycemia level was decreased in plasma glucose after 60 min of administration of 20 mg/kg of the extract, which was similar to the acarbose 50 mg/kg effect, while the oral administration of the extract (20 mg/kg) for 28 days in alloxan diabetic mice showed a more significant antidiabetic activity than that of acarbose [134].

The aqueous seed extract in a dosage of 1 gm/day given to streptozocin diabetic rats brought about a significant reduction in blood glucose levels in comparison to control rats, and restored the function of the liver and kidney and the balance of the oxidative stress condition with prolonged treatment [245].

A significant growth improvement, the restoration of hyperlipidemia, the restoration of relative kidney weight, the amelioration of the elevations in serum urea and creatinine levels, the reverse in the elevation of hepatic markers (ALT and AST), the improvement in the hepatic and renal antioxidant enzymes activity, an increase in GSH (glutathinone) levels and a good glycemic control all were biological effects reported in the streptozocin diabetic rats which received an aqueous suspension of *P. dactylifera* seeds 1 gm/kg [133].

These findings were explained by the extract’s ability to decrease the glycation of enzymes and to scavenge the free radicals. It is worth noting that *P. dactylifera* seeds have a significant amount of antioxidants, and constitute one of the highest sources of polyphenols, surpassing grapes, flaxseed, and nut seeds [133].

The antihyperglycemic effect of the oral administration of date seed extract combined with insulin was comparable to the seed extract administered alone in streptozocin diabetic rats [132].

Finally, *P. dactylifera* leaf ethanolic extract was investigated for its antidiabetic effects in alloxan diabetic rats. Their blood glucose level was checked after receiving different doses of 100, 200, and 400 mg/kg. The plasma insulin level was increased, and a significant reduction in blood glucose, serum triacylglycerol, and cholesterol were reported when compared with the control group [246].

### 4.30. Punica granatum (Punicaceae)

The treatment of streptozocin diabetic guinea pig or rats with fruit extract from pomegranate showed a marked hypoglycemic effect. It improved glucose homeostasis by restoring the level of glycogen, elevated hepatic glucose-6-phosphate dehydrogenase activity, improved total protein level, improved blood urea nitrogen and creatinine levels, reduced serum levels of total cholesterol and triglycerides, and enhanced the antioxidant activity in treated animals [247].

The mechanism for such effects was attributed to the active principles present in these extracts, such as polyphenols and flavonoids [247].

### 4.31. Rhus coriaria (Anacardiaceae)

Blood glucose and MDA (malondialdehyde) levels of the liver and kidney were lower in diabetic animals treated with aqueous extracts of R. coriaria, a source of antioxidant and free radical-scavenging activities [248].

Dogan et al. reported an antidiabetic property, with beneficial effects on lipid profile, levels of serum enzymes, renal function markers, *α*-glucosidase activity, and lipid peroxidation for lyophilized hydrophilic extract, obtained from the fruit in streptozocin diabetic rats, all without toxic effects [249].

### 4.32. Rheum turkestanicum (Polygonaceae)

The administration of the macerated extract significantly reduced the levels of fasting blood glucose and HbA1c, while the decoded extract did not change serum glucose over a three-week period, and these conflicting results can be explained by the differences between the type of extract and the duration of the study [250,251].

Generally, the antidiabetic effects of the *Rheum* genus may be related to the presence of high amounts of anthraquinones and flavonoids. They work through the regulating genes involved in the control of metabolism, enhancing the effect of insulin on glucose uptake, decreasing glucose absorption from the small intestine, enhancing glucose uptake in tissues, reducing gluconeogenesis, stimulating insulin release from *β*-cells, and preserving *β*-cell mass [250].

### 4.33. Salvia hypoleuca (Lamiaceae)

The effect of *S. hypoleuca* ethanolic extract was elucidated in normal and alloxan diabetic rats. In diabetic rats, blood glucose levels were significantly reduced after consumption of the extract, and the histological studies showed a restoration effect of the pancreatic islet cells [252].

*α*- amylase activity was reduced by *S. hydrangea* and various other species of the *Salvia* genus [253].

### 4.34. Salvia officinalis (Lamiaceae)

The methanolic extract of *S. officinalis* decreased the blood glucose level in alloxan diabetic rats in a dose-dependent manner, similarly to acarbose [166].

The administration of *S. officinalis* to diabetic rats lowered glucose, increased insulin level, and decreased lipid parameters [204].

The essential oil from sage increased hepatocyte sensitivity to insulin and inhibited gluconeogenesis. Rosmarinic acid and other phenolic components of the plant inhibited *α*-glucosidase. Altogether, sage may act by similar hypoglycemic mechanisms to metformin and acarbose [166].

### 4.35. Securigera securidaca (Fabaceae)

The oral administration of seed suspension significantly reduced elevated serum glucose concentration and increased antioxidant power in alloxan diabetic rats [254].

The capability of lowering serum glucose and lipid profile level using different organic solvents, carbon tetrachloride, 70% ethanol–water, and dichloromethane were investigated in an animal model [255].

Carbon tetrachloride solvent had the best and most significant activity for reducing glucose and lipid levels in comparison to other solvents, probably because of the higher content of sterols and fatty acids [255].

The hypoglycemic action of *S. securidaca* extract may be due to either enhanced insulin secretion, the inhibition of gluconeogenesis and direct stimulation of glycolysis, the reduction in serum glucagon level, or an extra-hepatic effect [176,255].

The hypoglycemic activity of methanol and chloroform fractions of *S. securidaca* seeds were comparable to glibenclamide. The higher doses of both extracts showed equal hypoglycemic responses to insulin. In addition, three cardiac glycosides were isolated as active constituents responsible for its hypoglycemic activity, and were found to act via an increase in insulin levels in a diabetic mouse model [256].

### 4.36. Solanum nigrum (Solanaceae)

The antidiabetic activity was reported for extracts of different parts of *S. nigrum* [257]. The administration of 1 g/L of *S. nigrum* fruit extract with the drinking water of diabetic animals decreased blood glucose levels. It was suggested that the plant can repair pancreatic *β*-cells, and enhance insulin secretion or increase glucose transporter translocation in the cell membrane, which then decreases blood glucose levels [257].

### 4.37. Teucrium polium (Lamiaceae)

The aqueous extract of *T. polium* showed a marked antihyperglycemic effect in diabetic rats. Previously, several studies reported hypoglycemic activity using different plant extracts, parts, and doses, and the ability to produce a dose-dependent stimulation of basal insulin release [258].

The plant had the ability to protect and in part restore the secretory function of *β*-cells in pancreatic tissue. Some flavonoids of the plant could inhibit hepatic gluconeogenesis, could exert an insulin-mimetic effect, and can attenuate oxidative stress and lipid peroxidation [259].

The added *T. polium* extract on the pancreatic MIN6 cell line caused changes in the *β*-cell cytosolic calcium ions concentration, triggered by the initiation of insulin release. Altogether, *T. polium* was thought of as a potential candidate with diabetic pharmacological modulation qualities [260].

### 4.38. Trigonella foenum-graecum (Fabaceae)

The antidiabetic activity of fenugreek has been studied the most, and has also been utilized by diabetic patients [261].

The daily oral administration of *Trigonella* seed-derived extract for type-2 diabetic rats decreased serum glucose, increased liver glycogen content, and enhanced total antioxidant status. In addition, in cultured 3T3-L1 adipocytes, glucose transport and insulin action were increased by *Trigonella* [262].

Fenugreek seed-derived polyphenols were observed to cause insulin-sensitizing actions, and were found to be comparable to metformin in the rat model [263]. A significant decrease in plasma glucose and insulin levels was seen in animals receiving a fructose-rich diet and aqueous extract of *Trigonella* [264].

Taking all this together, these studies suggest that the antidiabetic effect of fenugreek is mediated through the inhibition of carbohydrate digestion and the absorption and enhancement of peripheral insulin action [261].

### 4.39. Vaccinium arctostaphylos (Ericaceae)

The extract obtained from *V. arctostaphylos* berries produced a dose-dependent reduction in the *α*-amylase activity, and the compound responsible was Malvidin3-O-*β*-glucoside [265].

A dose-dependent antihyperglycemic effect of *V. arctostaphylos* fruit ethanolic extract was reported in alloxan diabetic rats. The effect was observed upon long-term treatment, and it was comparable to that of metformin. Besides that, the extract can improve the antioxidant and lipid profiles [266].

### 4.40. Vitex agnus-castus (Lamiaceae)

The hypoglycemic activity of *V. agnus-castus* was confirmed in streptozocin diabetic rats. The hypoglycemic and pancreatic protective effects were observed in aging animal models after treatment with fruit extract. Aging, insulin resistance, and *β*-cells destruction were all improved after treatment with the extract [267]. Glycation occurs due to high blood glucose levels for various structural and functional proteins. The fruit extract of *V. agnus-castus* showed the strongest antiglycation inhibitory activity, which may lead to a protective effect [268].

### 4.41. Urtica dioica (Utricaceae)

The antidiabetic activity of *U. dioica* extract was attributed to the insulin secretagogues effect and *α*-amylase and/or *α*-glucosidase inhibitory activity [269,270]. The hydro-distillate of the plant reduced fasting blood glucose and recovered the insulin levels in streptozocin diabetic rats—an effect not seen using glibenclamide. This effect can be attributed to the recovery of the pancreatic damage [269,270].

Ranjbari et al. reported a significant improvement in diabetic markers, increased insulin sensitivity, decreased insulin resistance, and improved the function of *β*-cells with the administration of *U. dioica* aqueous extract and swimming activity in diabetic rats [271]. Exposing the L6-GLUT4myc cell line to 125 and 250 μg/mL of nettle leaf and stem extracts doubled glucose transporter translocation to the plasma membrane, and increased the insulin-stimulated state 1.6-fold [272].

### 4.42. Zataria multiflora (Lamiaceae)

The essential oil of *Z. multiflora* reduced plasma glucose levels in experimental rats [273]. Antioxidant therapy is considered a major strategy for diabetes treatment. *Z. moltiflora* is an antioxidant source, with increased insulin level and reduced plasma glucose level. Recently, *α*-amylase inhibitory activity was reported for the plant [273,274].

### 4.43. Ziziphus vulgaris (Rhamnaceae)

The water extract of *Z. vulgaris* significantly decreased fasting blood glucose, cholesterol, and triglyceride levels after 14 days of treatment. The levels of HDL-cholesterol and insulin, and the activities of liver enzymes were not changed significantly in the extract-supplemented group compared to the control group [275].

Both the extract of *Z. spina-christi* leaves and its major saponin glycoside, christinin-A, improved the oral glucose tolerance and potentiated glucose-induced insulin release [276] (Table 8, Table 9 and Table 10).

## 5. Medicinal Plants with Potential Antidiabetic Activity in Iraq

The prevalence of type 2 diabetes in Iraq reached epidemic levels in 2007. Nowadays, around 1.4 million of Iraqis have diabetes [336,337].

### 5.1. Bauhinia variegate (Caesalpiniaceae)

The antihyperglycemic activity of the *B. variegate* was investigated in diabetic mice. The treatment resulted in a significant reduction effect for the blood glucose level compared to glibenclamide. B. variegate is rich in tannins, polyphenols and flavonoids, constituents that are reported to have antidiabetic effects [338].

### 5.2. Momordica charantia (Cucurbitaceae)

Over 140 studies worldwide investigated different extracts and ingredients of the plant for its antidiabetic activity and its mechanisms of action in both human and animal models [339].

The hypoglycemic effect was thought to be mediated through the stimulation of peripheral and skeletal muscle glucose utilization, the inhibition of intestinal glucose uptake, the inhibition of adipocyte differentiation, the suppression of key gluconeogenic enzymes, and the stimulation of key enzymes of the pentose phosphate pathway, and the preservation of islet *β*-cells and their functions [339,340].

### 5.3. Rheum ribes (Polygonaceae)

Previously, the hypoglycemic effect of different *R. ribes* extracts was reported. The aqueous root extract showed a significant hypoglycemic effect. In vitro, the extract stimulated insulin release from INS-1E cells. The hypoglycemic-active fraction contained anthraquinone glycosides of aloe-emodin, physcion, and chrysophanol derivatives [341].

Additionally, the aqueous root extract of *R. ribes* showed a hypoglycemic effect in a dose-dependent manner in alloxan diabetic rats [342].

*R. ribes* evoked an inhibitory effect for both *α*-amylase and *α*-glucosidase that showed a striking similarity to acarbose [343]. In another study, a significant increase in the activity of serum amylase in diabetic rats was observed after treatment with *R. ribes* [344].

The hypoglycemic effect of *R. ribes* may be due to the presence of insulin-like substances in plants that stimulate the regeneration and reactivation of *β*-cells to produce more insulin. Flavonoids may lead to the prevention or delaying of the progression of microvascular changes in the islet of Langerhans. However, these improvements did not lead to 100% recovery [344] (Table 11, Table 12 and Table 13). 

## 6. Medicinal Plants with Potential Antidiabetic Activity in Jordan

Over the past few decades, diabetes prevalence has increased to 13.1% according to the WHO.

### 6.1. Achillea santolina (Asteraceae)

The acute and chronic administration of aqueous extracts of *A. santolina* resulted in significant and marked reductions in serum glucose level in streptozocin diabetic rats [349].

An acute antihyperglycemic trend was observed for *A. santolina* bolus in starch-fed rats, explained by the intestinal luminal activation of effective entities. In addition, the treatment of MIN6 cells with the plant extract promoted dose-dependent pancreatic *β*-cell proliferation and monolayers expansion [343,350].

These effects may partly result from the inhibition of carbohydrates absorption. Moreover, *A. santolina* influenced insulin receptors or *β*-cells due to antioxidant activity [349].

### 6.2. Artemisia herba alba (Asteraceae)

The plant is a popular folk remedy for diabetes. The obtained data of several studies indicated that the aqueous, alcoholic, and organic solvents extracts of several parts of *A. herba alba* produced significant hypoglycemic effects in diabetic animals. The antidiabetic effect was comparable to that of the usual hypoglycemic drugs repaglinide, insulin, metformin, and glibenclamide [351,352].

It was hypothesized that the hypoglycemic effect is due to thujone, which can increase glucose transporter through the activation of adenosine monophosphate-activated protein kinase [352].

### 6.3. Artemisia sieberi (Asteracea)

The essential oil obtained from *A. sieberi* exhibited a significant reduction in blood glucose level in alloxan diabetic rats, which was comparable to metformin. The major constituents of the *Artemisia* species exert a potent antioxidant activity that provides a protective effect against diabetes [353].

A similar hypoglycemic effect was obtained in alloxan diabetic rabbits after treatment with aqueous extract from *A. sieberi*. The claimed mechanism could possibly be due to the increased peripheral glucose utilization, and the inhibition of the proximal tubular reabsorption mechanism for glucose in the kidneys [354].

### 6.4. Arum dioscoridis and palaestinum (Araceae)

A dual *α*-amylase- and *α*-glucosidase-inhibitory effect was observed for both Arum species as compared to acarbose. Apigenin, luteolin, and vitexin were identified in both species. All were strongly associated with the starch-digestive enzymes inhibitory effect [355]. The aqueous extract of *A. dioscoridis* significantly decreased the acute postprandial hyperglycemia induced in overnight fasting rats [355].

### 6.5. Crataegus aronia (Rosaceae)

The leaf extract of unripe fruits of *C. aronia* normalized plasma lipid peroxide levels and lowered blood glucose levels in diabetic animals [356]. The aqueous extracts of *C. aronia* at doses of 100, 200, and 400 mg/kg significantly decreased acute postprandial hyperglycemia and glycemic excursions in normoglycemic rats comparably to acarbose [357].

In addition, the *C. aronia* aerial parts’ and fruits’ aqueous extracts (0.1–10 mg/mL) gave rise to dual inhibitors of *α*-amylase and *α*-glucosidase [357].

A synergistic effect of antidiabetic and antioxidant activity was reported for the *C. aronia* extract of the whole plant. It was thought that the plant works through the inhibition of hepatic insulin resistance without affecting blood insulin level [155].

### 6.6. Cichorium pumilum (Asteraceae)

The ethanolic extract of *C. pumilum* leaves showed a significant antihyperglycemic activity [358].

### 6.7. Eryngium creticum (Apiaceae)

Leaf decoction was traditionally used as an antidiabetic therapy in Palestine and Jordan [359]. The aqueous extract of aerial parts decreased glucose levels in both diabetic and non-diabetic animals, but the effect was significant for the hyperglycemic rats only [358].

Although an acute antihyperglycemic property was observed after *E. creticum* bolus treatment in starch-fed rats, the in vitro inhibitory activity did not relate to the in vivo activity [360].

In another study, the MIN6 cell line was treated with an aqueous extract of *E. creticum*. It significantly potentiated glucose-stimulated insulin secretion, and augmented the *β*-cell proliferation in a dose-dependent manner (0.01 mg/mL) [350].

### 6.8. Geranium graveolens (Geraniaceae)

*G. graveolens* can improve glycemic control via *α*-amylase and *α*-glucosidase enzyme inhibition, and this was confirmed through acute and potent antihyperglycemic trends in starch-fed normal rats [360].

After treatment with the extract, the pancreatic mass increased maximally. This effect can provide a promising avenue for *β*-cell death in diabetes care [361].

Additionally, the administration of essential oil from *G. graveolens* significantly decreased blood glucose level and restored perturbed antioxidant activity comparably to glibenclamide activity in alloxan diabetic rats [362].

### 6.9. Phaseolus vulgaris (Fabaceae)

The aqueous seed extract and the alcoholic fresh pod extract of *P. vulgaris* were active for hypoglycemic potential. The antihyperglycemic activity might involve the inhibition of *α*-amylase activity and the enhancement of the activity of insulin-producing *β*-cells [358,363].

### 6.10. Pistacia atlantica (Anacardiaceae)

The administration of hexane seed extract decreased the blood glucose level to normal in diabetic rats. In addition, the aqueous leaf extract inhibited *α*-amylase and *α*-glucosidase enzymes comparably to metformin and glipizide in starch-fed rats [343].

Previously, it was found that *P. atlantica* can improve glucose homeostasis via insulin secretagogue and glucose absorption restrictive activity [350].

### 6.11. Tecoma stans (Bignoniaceae)

The ethanolic leaf extract of *T. stans* showed an antihyperglycemic action in alloxan diabetic rats, and this result was in agreement with previous studies that examined such effects in insulin-sensitive and insulin-resistant mice, cultured human adipocytes, and streptozocin diabetic rats, an effect attributed to a denominated alkaloid tecomine and tecostatine [358].

### 6.12. Teucrium polium (Lamiaceae)

In 1987 the hypoglycemic activity of *T. polium* was reported. The aqueous, hydroalcoholic extracts of aerial parts and leaves caused a significant reduction in blood glucose concentration in normoglycemic and hyperglycemic rats [258,358].

It seemed that the *T. polium* extract had an insulinotropic property, was able to enhance insulin secretion, and was capable of regenerating the islets of Langerhans [258,364].

### 6.13. Varthemia iphionoides (Asteraceae)

Since 1997, the aqueous extract of *V. iphionoides* has been reported to exhibit a hypoglycemic activity in both normal and diabetic rats [365].

Recently, a dose-dependent dual *α*-amylase and *α*-glucosidase inhibitory activity was demonstrated. In addition, a pancreatic proliferative capacity of *V. iphionoides* extracts in chronic treatments was identified [361] (Table 14, Table 15 and Table 16).

## 7. Medicinal Plants with Potential Antidiabetic Activity in Lebanon

### 7.1. Centaurea horrida (Asteraceae)

The root extract of *C. horrida* showed a significant drop in blood glucose level using several doses; therefore, the dose of 100 mg/kg was the most effective the for acute and sub-acute blood glucose of hyperglycemic mice [381].

The extract might inhibit endogenous glucose production or intestinal glucose absorption [381].

### 7.2. Hordeum spontaneum (Poaceae)

Barley is one of the most ancient crops cultivated for food. The importance of barley as a nutraceutical ingredient has increased since the high content of *β*-glucan affects glycemic control [381,382].

Hordeum spontaneum grain ethanolic extract reduced the blood glucose level in hyperglycemic mice, and was more potent than glibenclamide [381].

### 7.3. Inula viscosa and Inula vulgaris (Asteraceae)

The isolated flavanone 7-O-methylaromadendrin stimulated glucose uptake and improved insulin resistance [383].

Remarkable acute and subacute effects on blood glucose levels were reported for *I. viscosa* and *I. vulgaris* alcoholic extracts used in alloxan diabetic rats. The effect of *I. viscosa* was higher in comparison to *I. vulgaris* [384].

The diabetic mice treated with 25 mg/kg of each extract exhibited a significant rise in serum catalase activity—an enzyme of the antioxidant defense mechanisms that decreases oxidative stress. It was reported that the long-term treatment of diabetes with the two plant extracts reversed the activities of the free radicals [384].

### 7.4. Psoralea bituminosa (Fabaceae)

The aqueous extract of *P. bituminosa* leaves lowered blood glucose levels in streptozocin diabetic rats [385].

The acute effect of ethyl acetate, hexane, and ethanol in alloxan diabetic rats was investigated. At all doses of 25, 50, and 100 mg/kg, a significant decrease in blood glucose level and a significant antioxidant activity was reported [386].

### 7.5. Salvia libanotica (Lamiaceae)

The *Salvia fruticosa* leaves extract caused a statistically significant reduction in blood glucose level in alloxan hyperglycemic rabbits [381].

*Salvia libanotica fruticosa* root extract at several doses showed a significant drop in blood glucose level in hyperglycemic mice. The mechanism of the hypoglycemic activity may be due to the inhibition of the endogenous glucose production, or the inhibition of intestinal glucose absorption [381] (Table 17, Table 18 and Table 19).

## 8. Medicinal Plants with Potential Antidiabetic Activity in Palestine

### 8.1. Atriplex halimus (Chenopodiaceae)

Saltbush was an extremely effective antidiabetic plant with an insulin-potentiating property in an animal model for diabetogenesis [392].

The aqueous extract of *A. halium* was beneficial in reducing the elevated blood glucose level and hepatic levels in streptozocin diabetic rats with a toxic effect at a dose of 200 mg/kg [392].

*A. halimus’* antidiabetic activity was mediated at least through increasing glucose uptake to the muscle, liver, and fat cells [272].

### 8.2. Ocimum basilicum (Lamiaceae)

A significant dose-dependent inhibition against intestinal sucrose, maltose, and pancreatic *α*- amylase was reported [125].

The extract of aerial parts of O.basilicum showed a higher hypoglycemic effect in treated rats than in metformin-treated rats. It improved oral glucose tolerance, increased the liver glycogen content in a dose-dependent manner, and enhanced glucose mobilization by stimulating hepatic glycogen synthesis [393].

Methanol, hexane, and dichloromethane extracts of the aerial parts of *O. basilicum* were evaluated for their antidiabetic properties in vitro. They increased glucose transporter translocation to the plasma membrane, especially the methanol and hexane extracts [394].

### 8.3. Sarcopoterium spinosum (Rosaceae)

An aqueous extract of *S. spinosum* was used to treat multiple cell lines (RINm pancreatic *β*-cells, L6 myotubes, 3T3-L1 adipocytes, and AML-12 hepatocytes). Skeletal muscle, adipose tissue, and hepatocytes have been reported to have insulin-like effects [395].

The plant extract decreased insulin resistance and increased insulin sensitivity in diabetic mice in another study [396].

The extract of the aerial parts inhibited *α*-amylase and *α*-glucosidase, also enhancing glucose cell uptake by activating the insulin-signaling cascade [397].

Glucose tolerance and insulin sensitivity were improved in high-fat diet mice and diabetic mice—two models of insulin resistance—and insulin resistance was decreased after treatment with *S. spinosum* [398].

### 8.4. Trigonella foenum-graecum (Fabaceae)

Fenugreek is well-known as an antidiabetic remedy. Preliminary animal and human trials proposed the possible hypoglycemic and antihyperlipidemic properties of fenugreek powder, seed, leaf and their extracts [399,400].

Fenugreek contains galactomannan-rich soluble fiber, which combines with bile acid and lowers triglyceride and LDL cholesterol levels, and may be responsible for the antidiabetic activity of the seeds. In addition, it contains nicotinic acid, alkaloid trigonelline, and coumarin, which were proven to be responsible for its antidiabetic properties [272,399].

The extract of fenugreek was able to inhibit *α*-amylase activity in a dose-dependent manner, suggesting that the hypoglycemic effect of the used plant extract was mediated through insulin mimetic properties [400].

The seed extract of *T. foenum-graecum* doubled the translocation of glucose transporter to plasma membrane, and at non-cytotoxic concentrations [272].

Finally, studies showed that the treatment of severely diabetic rabbits with a fenugreek seeds isolated compound corrected the altered serum lipids, tissue lipids, glycogen, enzymes of glycolysis, gluconeogenesis, glycogen metabolism, polyol pathway and antioxidant enzymes, and corrected all the histopathological abnormalities seen in the pancreas, liver, heart and kidneys [401].

### 8.5. Withania somnifera (Solanaceae)

The aqueous extract of *W. somnifera* significantly improved insulin sensitivity index in streptozocin diabetic rats [402]. The leaf and root extracts of *W. somnifera* increased glucose uptake in myotubes and adipocytes in a dose-dependent manner, and increased insulin secretion. Withaferin was found to increase glucose cell uptake [403].

The extract also decreased blood glucose levels in alloxan and streptozocin diabetic rats [402,404]. The antidiabetic activity may be due to an increase in hepatic metabolism, increased insulin release from pancreatic *β*-cells, or the insulin-sparing effect [405] (Table 20, Table 21 and Table 22).

## 9. Medicinal Plants with Potential Antidiabetic Activity in Turkey

Turkey is considered as one of the richest countries in the temperate world with regards to the floristic diversity [419].

*Phlomis armeniaca*, *Salvia limbata* and *Plantago lanceolata* exhibited weak inhibitory activity against *α*-amylase and pronounced inhibitory activity against *α*-glucosidase [419].

Methanolic and aqueous extracts of *Hedysarum varium* and *Onobrychis hypargyrea* showed significant inhibition against *α*-glucosidase when compared to acarbose [420].

In another study, *Hedysarum varium*, *Onobrychis hypargyrea* and *Salvia limbata* extracts showed inhibition against *α*-amylase and potent inhibition against *α*-glucosidase compared to acarbose [420].

The hydroalcoholic extract of *Helichrysum graveolens* inhibited the *α*-amylase enzyme similarly to acarbose at a dose of 3000 μg/mL. Several secondary metabolites were reported in this genus, and chlorogenic acid was found to be the major phenolic in the extract [421].

The lyophilized hydrophilic extract obtained from leaves of *Cichorium intybus* effectively suppressed the activity of *α*-glucosidase, but the inhibitory activity towards *α*-amylase was limited. It is well known that plant extracts which selectively inhibit *α*-glucosidase are the preferred agents in controlling the uptake of glucose in diabetes. The inhibition of both enzymes would result in abnormal bacterial fermentation in the colon due to the presence of undigested carbohydrates [422].

*Cinnamomum verum* aqueous and ethanolic extracts of the bark showed more affinity for *α*-amylase than for *α*-glycosidase enzyme. It was reported that cinnamon might improve anthropometric parameters, and can also activate the glycogen synthase through stimulating glucose uptake, and inhibiting glycogen synthase kinase [423].

Another investigation revealed that the methanol extract of *Origanum onites* exerted a higher inhibition against *α*-amylase and *α*-glucosidase compared to the aqueous extract [424].

Additionally, the aqueous and methanol extracts of *Mentha pulegium* had more affinity for *α*-amylase than for the *α*-glycosidase enzyme. Both caused greater inhibition of *α*-glycosidase than acarbose [425].

Both ethyl acetate and petrolum ether extracts of *Haplophyllum myrtifolium* showed a similar, significant, *α*-amylase inhibition rate that was higher than that of the methanol and water extracts. For the *α*-glucosidase-inhibitory effect, the ethyl acetate extract exhibited the highest activity followed, by the petroleum ether and methanol extracts. However, the water extract tested did not have any effect on the *α*-glucosidase activity [426].

Previous studies showed that the extracts rich in phenolics possessed good inhibitory activity for *α*-glucosidase and *α*-amylase. Interestingly, the ethyl acetate extract had the highest level of phenolics [426].

*Laurus nobilis* essential oil and its major component, 1,8-cineole, inhibited *α*-glucosidase by competitive inhibition, but 1-(*S*)-*α*-pinene and *R*-(+)-limonene were uncompetitive inhibitors. All possessed an in vitro antioxidant property [427] (Table 23).

### 9.1. Cistus laurifolius (Cistaceae)

The ethanolic extract of the leaves was found to have a promising antidiabetic activity in streptozocin diabetic rats and significant inhibitory activity for *α*-amylase and glucosidase enzymes [437].

Twelve major flavonoids and other phenolic compounds were isolated from the ethanolic extract of the leaves. Some were reported to inhibit aldose reductases, enhance both basal and insulin-stimulated glucose uptake, prevent *β*-cell apoptosis and modulate their proliferation, and inhibit *α*-glucosidase and amylase enzymes [437].

### 9.2. Juniperus oxycedrus (Cupressaceae)

Infusions and decoctions of *Juniperus* fruits and leaves were used for diabetes in Turkey as folk remedy. According to the literature, different species of *Juniper* possess hypoglycemic effects [438,439].

Aqueous and ethanolic extracts of *J. oxycedrus* were tested in streptozocin diabetic rats. The aqueous extract exhibited a mild hypoglycemic effect, wherein the ethanolic extract showed a higher and more continuous hypoglycemic effect. In the same study, isolated shikimic acid possessed a promising antidiabetic activity at a dose of 30 mg/kg [438].

Additionally, the effects of *J. oxycedrus* leaf extracts were investigated in diabetic animals. The study indicated that the treatment of diabetic rats with the aqueous extract showed a moderate hypoglycemic effect, while the antihyperglycemic effect of ethanol extract was significant and continuous at a dose of 1000 mg/kg [440].

Leaf and fruit hydroethanolic extracts decreased blood glucose level and lipid peroxidation in tissues remarkably [441].

Phytochemical analysis showed that *J. oxycedrus* was rich in secondary metabolites that were responsible for its antidiabetic effect [441].

The increase in insulin stimulation, insulin sensitivity and release were related to the unsaturated fatty acids-rich fraction of leaves extract [440].

Shikimic acid, a major molecule found in berry extract, has a well-known antidiabetic activity through its antioxidant activity and ability to increase the insulin-like growth factor-1 [438].

Augmenting Zn concentration in the liver is another possible mechanism. Zn plays a key role in the synthesis and action of insulin both physiologically and in the pathologic state of diabetes [441].

### 9.3. Heracleum persicum (Apiaceae)

The increased levels of plasma glucose in diabetic rats were lowered by extract supplementations in all groups after 21 days. The antihyperglycemic action resulted from the potentiation of insulin release from existing *β*-cells, and regenerating the functions of islet cells over time [320].

Among the isolated compounds taken from the roots of *H. persicum*, pimpinellin-dimer was identified as a potent *α*-glucosidase inhibitor, more effective than acarbose. In addition, it exhibited significant antioxidant activity [442].

It is well known that oxidative stress is caused by reactive oxygen species (ROS) and plays an important role in the pathogenesis of various degenerative diseases, such as diabetes; similarly, postprandial oxidative stress is associated with a higher risk for diabetes, and therefore, *H. persicum* can be used as a potent antioxidant and as a strong inhibitor of *α*-amylase in the improvement of type 2 diabetes patients [443].

### 9.4. Juniperus foetidissima and Juniperus sabina (Cupressaceae)

All extracts of *J. foetidissima* and *Sabina* had promising *α*-glucosidase inhibitory activities. The ethanolic leaf extract of *J. foetidissima* had the highest activity. On the other hand, the *α*-amylase inhibitory activity of *Juniperus* extracts was moderate [439].

In the same study, the ethanolic fruit extract of *J. foetidissima* at a dose of 500 mg/kg significantly lowered the blood glucose level for diabetic animals, similarly to glipizide [439].

The antidiabetic activity was attributed to amentoflavone, which showed significant inhibitory activity for carbohydrate digestive enzymes, as well as positive effects on insulin resistance [439].

### 9.5. Origanum minutiflorum (Labmiaceae)

A significant time- and dose-dependent antihyperglycemic activity of aqueous *O. minutiflorum* extract was observed in diabetic rats. This effect was attributed to its flavonoid-rich contents, well-known compounds for their antidiabetic effects [444].

### 9.6. Salvia triloba (Lamiaceae)

The methanolic extract of *S. triloba* decreased blood glucose level in a non-dose-dependent manner in streptozocin/nicotinamide diabetic rats. It performed its action by improving insulin sensitivity [445].

### 9.7. Thymus praecox (Lamiaceae)

The methanolic extract of *T. praecox* decreased blood glucose level in Streptozocin/nicotinamide diabetic rats. It performed its action by improving insulin sensitivity [445] (Table 24, Table 25 and Table 26).

## 10. Middle East Statistics

The flora in the Middle East area provides diverse useful species. A total of 140 medicinal plant species were studied scientifically for their potential antidiabetic activities. These species were classified among 59 families (Figure 1). Lamiaceae (20 species, 14%) and Asteraceae (16 species, 11%) have a greater number of medicinal plants with antidiabetic activities in Middle East countries.

Scientific investigations try to validate the antidiabetic potential of several Middle East medicinal plants using several methods of extraction, plant organs, and solvents. The majority of the studies used aqueous maceration as an extraction method (Figure 2 and Figure 3) Water was mostly used as a solvent as the researchers simulated the ethnobotanical use by people. Among plant parts, leaves were the most frequently used part (28%) (Figure 4.)

The prevalence of type 2 diabetes has risen dramatically. About 422 million people worldwide have diabetes, particularly in low- and middle-income countries, and 1.6 million deaths are directly attributed to diabetes each year (World Health Organization, 2020).

There are numerous plant species in the Middle East. They were estimated to exceed 13,500 species, mainly found in Turkey and Iran [456]. To exert their effects, plants with potential antidiabetic properties have many targets and diverse mechanisms of actions, (Figure 5).

## 11. Conclusions

Above is the collected information regarding plants used in the treatment of diabetes mellitus in Middle East countries. In recent years, the ethnobotanical and traditional use of natural compounds, especially those of plant origin, has received much attention, as they are well tested for their efficacy and are generally believed to be safe for the treatment of human ailments. It is the best classical approach in the search for new molecules for the management of various diseases.

Many new bioactive drugs isolated from plants showed antidiabetic activity that was equally as potent as, and sometimes even more potent than, known oral hypoglycemic agents, such as metformin and glibenclamide. However, many other active agents obtained from plants have not been well characterized. More investigations must be carried out to elucidate the mechanisms of action and the toxicity of medicinal plants with antidiabetic effects.

This review contributes scientifically to evidence for the ethnobotanical use of medicinal plants as antidiabetic agents. Work has to be done to define the target, mechanism of action and the responsible compound for activity. In addition, safety and pharmacokinetic parameters should be investigated.

## Figures and Tables

**Figure 1 molecules-26-00742-f001:**
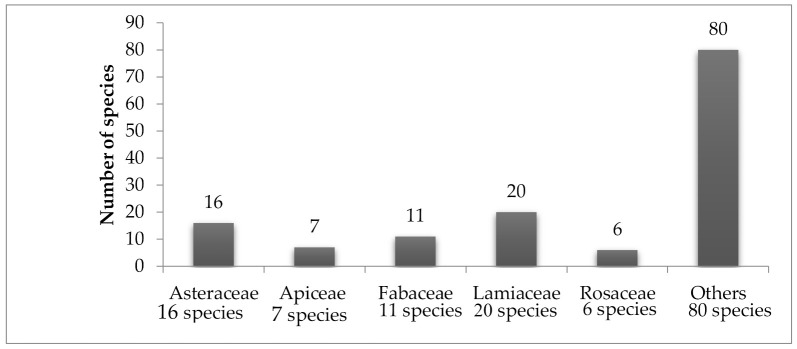
Botanical families studied.

**Figure 2 molecules-26-00742-f002:**
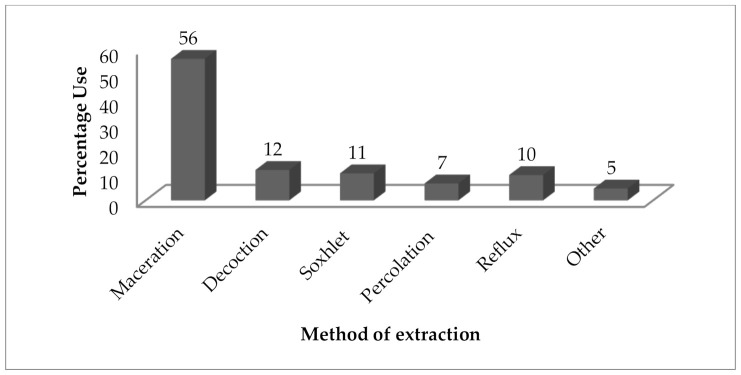
The percentage of extraction methods.

**Figure 3 molecules-26-00742-f003:**
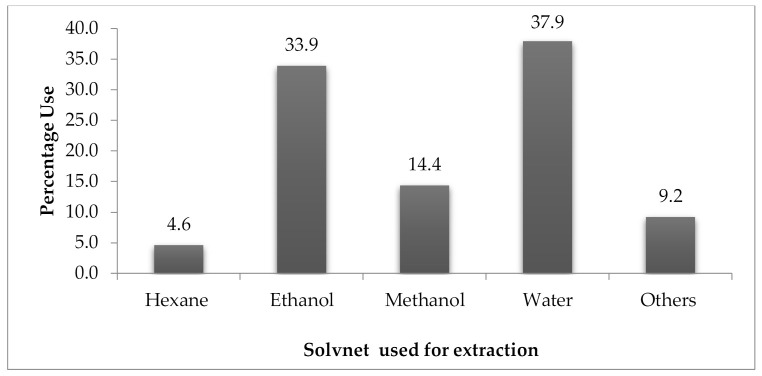
The percentage of solvents used for extraction.

**Figure 4 molecules-26-00742-f004:**
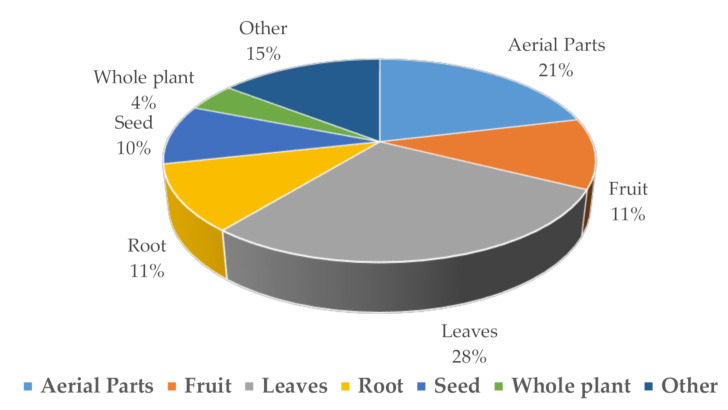
The percentage of different plant parts used.

**Figure 5 molecules-26-00742-f005:**
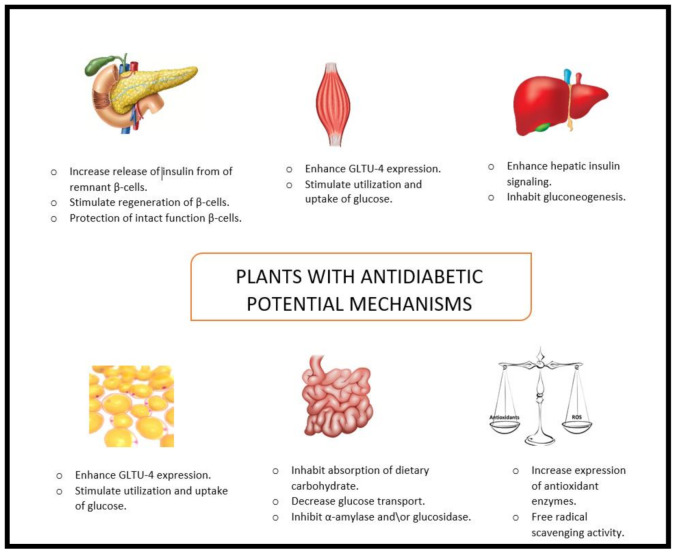
Targets of medicinal plant with antidiabetic activity.

**Table 1 molecules-26-00742-t001:** Medicinal plants of Arabian Peninsula ethnobotany.

County	Scientific Name	Common Name	Family	Traditional Use	Part Used	Reference
Oman	*Ajuga iva*	Chendgoura	Lamiaceae	Anthelmintic, analgesia, diuretic agent, diabetes	Leaf, stem	[71,72]
	*Moringa peregrina*	Shua	Moringaceae	Convulsions or infantile paralysis, diabetes	Pod oil, seed	[17,73]
	*Rhayza stricta*	Harmal	Apocynaceae	Bad breath, chest pain, conjunctivitis, constipation, diabetes, fever, skin rash, anthelmintic, increase lactation	Seed, whole plant	[74]
Qatar	*Cynomorium coccineum*	Tarthuth	Cynomoriaceae	The roots are edible and were sold in earlier times as a vegetable (Qatar) used as an aphrodisiac in Bahrain	Flower, root	[75]
Saudi Arabia	*Avicennia marina*	Shoura	Avicenniaceae	Smallpox, sores, pruritic, induce women infertility, diabetes	Branches	[76]
	*Caralluma sinaica*	DedElkalba	Asclepiadaceae	Anticancer, diabetes, leprosy, obesity, rheumatism	Leaf	[77,78,79]
	*Ducrosia anethifolia*	Not mentioned	Apiaceae	Analgesic, anxiety, backache, carminative, colic, cold, galactogogue, insomnia, pain reliever for headache, useful for irregularities of menstruation (Iranian folk medicine)	Aerial parts, seed, whole herb	[80]
	*Jatropha curcas*	Kharat	Euphorbiaceae	Ailments related to skin, cancer, digestive disease, diabetes, infectious disease, respiratory disease	Barkk, fruit, latex, leaf, root, seed	[79,81]
	*Loranthus acaciae*	Not mentioned	Loranthaceae	Antipyretic, cancer, colds, cosmetic, ear aches, headache, hypertension, mosquito Repellent, obesity, rheumatoid diseases, skin infections	Leaf	[82,83]
	*Lycium shawii*	Awsaj	Solonaceae	Diabetes, hypotensive agent	Flower, shoot	[39,84]
	*Marrubiumvulgare*	Not mentioned	Lamiaceae	Return to reference	Leaves, flowers, stem	[85]
	*Moringa oleifera*	Getha	Moringaceae	Diabetes, ascites, splenic enlargement, inflammatory swellings, abdominal tumors, colic, dyspepsia, fever, ulcers, paralysis, lumbago, skin diseases	Fruit	[86]
	*Ocimum forskolei*	Basil	Lamiaceae	Cosmetic, digestive agent, spasm, mosquito’s repellent factor, relieves fever, skin infections	Leaf	[45,87]
Saudi Arabia	*Plicosepalus curviflorus*	EnamElTalh	Loranthaceae	Cancer, diabetes	Stem	[46]
	*Retama raetam*	Al-retem	Fabaceae	Diabetes	Fruit	[88]
	*Rhizophora mucronata*	Kindale	Rhizosphoraceae	Elephantiasis, hematoma, hepatitis, ulcers, diarrhea or gastric motility disorder, diabetes	Bark, branches, flower, fruit, leaf	[76,89]
	*Salvadora persica*	Miswak	Salvadoraceae	Oral and dental cleaning tool	Bark	[49]
Yemen	*Azadirachta indica*	Neem	Meliaceae	Diabetes, GIT disorder, general health promoter, leprosy, respiratory disorders	Leaf, bark	[90]
	*Boswellia carterii*	Olibanum	Burseraceae	Anti-inflammatory, bruises, infected sores psychoactive, reduce the loss of blood in the urine from schistosomiasis infestation, tranquilizer, wound healing, various non-pharmaceutical applications	Resin	[58,91]
	*Cissus rotundifolia*	Arabian wax	Vitaceae	Appetizer, antipyretic carbuncles, dengue fever, malaria, rheumatic pain, snake bites	Leaf, root, stem,	[58]
	*Dracaena cinnabari*	Damm Al-Khwain	Agavaceae	Burn, control of bleeding, diarrhea, fractures, fevers, healing of fractures, hemorrhage, pain, stimulation of circulation, sprains, ulcers, wounds	Resin	[63,92]
	*Opuntia ficus-indica*	Cactus	Cactaceae	Allergy, analgesic, anti-inflammatory, dyspnoea, fatigue, glaucoma, gastric ulcer, health supporting nutrient, healing wounds, hypoglycemic agent, liver diseases, prostate cancer, urological problems	Flower, fruit	[93]
Yemen	*Pulicaria inuloides*	False fleabane	Asteraceae	Anthelmintic, carminative, cold, diuretic, fever, inflammation, insect repellent, pain, intestinal disorder, pyritic conditions in urogenetic organs	Flowers	[94,95]
	*Solenostemma argel*	Argel	Asclepiadaceae	Diabetes, cardiovascular disorders, gastrointestinal problems, kidney and liver diseases, pain, respiratory tract infections, urinary tract infections	Bark, leaf, stem	[69,96]

**Table 2 molecules-26-00742-t002:** Medicinal plants of Arabian Peninsula reported constituents use.

Country	Name	Chemical Constituents	Scientific Reports	Reference
Oman	*Ajuga iva*	Anthocyanins, ecdysteroids, flavones, glycosides, terpenes; diterpenes, triterpenes, tannins, withanolides	Antibacterial, antifungal, antihypertensive, antimycobacterial, antiplasmodial, hypoglycaemic, larvae and insect antifeedant activity	[12,16,97,98]
	Moringa	Flavonoid, isothiocyanate, glycosides, phytosterol, polyphenol, triterpene, volatile oil	Anticancer, antioxidant, antimicrobial, antidiabetic, anti-inflammatory, anti-spasmodic, hypertension, hepatotoxicity, lipid lowering activity, memory disorders	[17]
	*Rhazya stricta*	Indole alkaloid, flavonoids, carbolines, alkaloids with *β*-carboline nucleus (akuammidine, rhazinilam tetrahydrosecamine), triterpenes, tannins, volatile bases	Antioxidant, blood pressure, diabetes mellitus, immunomodulation effect	[22,99]
Qatar	*C. coccineum*	Flavonoids, organic acids, scharrides, lipids	Anticancer, antidiabetic, antioxidant, anti-tyrosinase antimicrobial, cardioprotective, immunity-improving, neuroprotective, increase and folliculogenesis, testicular development, spermatogenesis	[100,101]
Saudi Arabia	*Avicennia marina*	Alkaloids, glycosides, flavonoids, phenols, saponins, tannins, terpenoids	Antidiabetic, anti-inflammatory, antimicrobial, antioxidant, antiviral	[102]
	*Caralluma sinaica*	Coumarin, flavonoids, glycosides, phenolic alkaloids, steroids, tannins	Antidiabetic agent, antimicrobial	[78]
	*Ducrosia anethifolia*	Monoterpenes hydrocarbons(essential oil), coumarins	Anticancer, antidiabetic, anti-inflammatory, antimicrobial, anxiolytic, radical scavenging	[80,103]
Saudi Arabia	*Jatropha curcas*	C yclic peptide alkaloids, flavonoid, lignans, saponins, terpenes, diterpenoids	Antioxidant, anticancer, anti-inflammatory, gastroprotective, anthelmintic, antidiarrhea activity, antiulcer	[103,104,105,106]
	*Loranthus acaciae*	Alkaloid, cardiac glycosides, flavonoids saponins tannins, terpenoids, catechin	Analgesic, antidiabetic, anti-inflammatory, antimicrobial, antitumor, antioxidant, antihepatotoxic, antiviral	[38,107]
	*Lycium shawii*	Alkaloids, flavonoids, phenolics, tannins, saponins, glycosides, terpenoids, steroids, coumarins	Antioxidant, diuretic, laxative, anticancer, antimicrobial anti-inflammatory, tonic agent	[108]
	*Marrubiumvulgare*	Flavonoid, phenylpropanoids, terpenes, sesqui and diterpenes	Antihypertensive, analgesic, anti-inflammatory, antioxidant	[85]
	*Moringa oleifera*	Carotenoids, glucosinolates, flavonoids, phenolic acids, polyunsaturated fatty acids, tocopherols	Antidyslipidemic, anthelmintic, antihyperglycemic, anti-inflammatory, antimicrobial, antioxidant, apoptotic properties, antiproliferative, anti-ulcer, antiurolithiatic, hepatoprotective	[109]
	*Morus nigra*	Alkaloids, flavonoids, phenols	Antihyperlipidemic, antidepressant, antioxidant, neuroprotective	[110]
Saudi Arabia	*Ocimum forskolei*	Alkaloids, essential oils, flavonoids, glycosides, phenylpropanoids, saponins, steroids, tannins	Activities against bacteria and dermatophytes, antihyperglycemic, nematicidal activity, weak antioxidant	[87,111]
	*Plicosepalus curviflorus*	Flavonol rhamnosides, sesquiterpene lactones	Antihepatotoxic, antidiabetic, cytotoxic activities	[112]
	*Retama raetam*	Flavonoid, quinolizidine alkaloids	Antibacterial, antihyperlipidemic, antihypertensive, antioxidant, diuretic, hypoglycemic.	[48]
	*Rhizophora mucronata*	Alkaloids, anthocyanidins, anthraquinone, carotenoids, catechin, flavonoids, phenolics, saponin, steroids, triterpene	Antimicrobial, antioxidant, hepatoprotective	[89,102,113]
	*Salvadora persica*	Essential oil, organosulphur compounds, saponin, *β*-sitosterol	Anticonvulsant, antifertility, antimicrobial, analgesic, antiplaque, aphrodisiac, antipyretic, antiulcer, astringent, diuretic, hypolipidemic, stomachic activities.	[49,114]
Yemen	*Azadirachta indica*	Alkaloid, aromatic compound, flavonoid, flavone, isoprenoidssesquiterpenes, terpenes;triterpenoids tetraanortriterpenoid, saponins, limonoids	Antibacterial, antidiabetic, antifertility, antihypercholesteremic. antimalarial, antioxidant, antiulcer, anti-tumor	[9,58]
	*Boswellia carterii*	Pentacyclic triterpenoid, volatile oil	Anti-inflammatory, neuroprotective.	[115,116,117]
Yemen	*Cissus rotundifolia*	Steroids, flavonoids, *β*-sitosterol	Analgesic, antidiabetic, anti-inflammatory, antiparasitic, antiulcerative, antioxidant, hepatoprotective activity	[62,118,119]
	*Dracaena cinnabari*	Flavone, chalcone	Antidiarrheal, anti-hemorrhagic, anti-inflammatory antimicrobial, antiviral activity, antitumor activity, antiulcer, immunomodulatory, antioxidant	[92]
	*Opuntia ficus-indica*	Flavonoid glycoside, oxygenated monoterpenes, sesquiterpene hydrocarbons, aldehydes with non-terpenic structure	Antioxidant, antiulcer, cardioprotective, hepatoprotective	[120,121]
	*Pulicaria inuloides*	Essential oil, monoterpenes, diterpenes, sesquiterpene lactones and caryophyllane derivatives	Antimicrobial, antifungal, antimalaria, insecticides properties	[94,95]
	*Solenostemma argel*	Phenolic acids, flavones, glycosylated flavonoids, polyphenols, *β*-carotene, *β*-sitosterol, terpenes, mono and triterpenes	Antibacterial, antifungal, antioxidant	[69,96,122,123]

**Table 3 molecules-26-00742-t003:** Medicinal plants with potential antidiabetic activity in Arabian Peninsula.

Scientific Name	Part Used	Extraction Method, Solvent	Target	Intervention and Duration	Observations	Ref.
*Ajuga iva*	Root, stem, leaf	Ethanol, maceration	Streptozocin diabetic rats	21 days	Hypoglycemic activity	[25]
*Moringa pergrina*	Leaf	Chloroform, soxhlet	Mice	1 mL of 0.5%, 1%, 1.5%, or 2%, 21 days	Weight reduction, influenced the reproductive system	[21]
*Pteropyrum scoparium*	Leaf, root, stem	Ethanol, maceration	streptozocin diabetic rats	21 days	Hypoglycemic activity	[25]
*Rhazya stricta*	Root, stem, leaf	Ethanol, maceration	Streptozocin diabetic rats	21 days	Hypoglycemic activity	[25]
*Cynomorium coccineum*	Inner flesh of stem	Water, reflux	In vitro		Antioxidant activity associated with angiotensin converting enzyme	[27]
*Avicennia marina*	Leaf	Ethanol, maceration	Streptozocin diabetic rats	2 mg/gm, 4 weeks	Protected liver, improved the neurobehavioral changes, decreased inflammatory cells aggregation, vacuolation, and hemorrhage.	[30]
	Leaf	Water, decoction	Streptozocin diabetic rats	400 mg/kg, 6 weeks	Significant increase in the muscle levels of CAT, SOD and glutathione	[31]
*Caralluma sinaica*	Aerial parts, root	Ethanol (80%), maceration	Streptozocin diabetic rats	100 mg/kg, 30 days	Reversed streptozocin effect on glycogen content	[33]
*Ducrosia anethifolia*	Aerial parts	Ethanol (80%),maceration	Streptozocin diabetic rats	500 mg/kg, 45 days	Normalized liver enzymes, total protein, lipid, cholesterol levels, antioxidant markers, glycolytic, and elevated level of kidney biomarkers	[34]
*Jatropha curcas*	Aerial parts	Ethanol (96%), chloroform, hexane, maceration	Alloxan diabetic mice	400 mg/kg, 1 day	Safe up to dose of 5 g/kg	[124]
*Loranthus acaciae*	Leaf, stem	Ethanol, soxhlet	Alloxan diabetic rats, normal rats	500 mg/kg, 1 week	LD50 of the extract and its fractions was more than 5 g/kg, a potent anti-inflammatory and antioxidant effect was detected for the chloroform fraction	[38]
*Lycium shawii*	Aerial parts	Ethanol, decoction	Alloxan diabetic rats, normal rats	500 mg/kg, o.p or i.p, 90 days	None of the mice died up to 3 g/kg dose. Prolonged *L. shawii* treatment is toxic.	[39]
*Marrubium vulgare*	Aerial parts	Methanol, maceration	Streptozocin diabetic rats	500 mg/kg, 28 days	Significant reduction in plasma TC, TG, LDL, increased HDL	[40]
*Moringa oleifera*	Seed	Powder	Streptozocin diabetic rats	50, 100 mg/kg, 4 weeks	Restoration of the normal histology of both kidney and pancreas	[42]
*Morus nigra*	Leaf	Ethylalcohol, maceration	Streptozocin diabetic rats	500 mg/kg, 10 days	Significant increase in insulin level	[43]
*Ocimum basilicum*	Leaf	Water, infusion	In vitro	20, 18.2, 16.3, 14.5 mg/mL	Significant dose-dependent inhibition of sucrase, maltase and pancreatic *α*-amylase	[125]
*Ocimum forskolei*	Leaf, stem	Methanol (70%), maceration	In vitro	10, 20, 30 mg/mL	The inhibitory activity (IC_50_) of both leaf and stems methanol extracts are almost the same	[45]
*Plicosepalus curviflorus*	Leaf	Serial exhaustive extraction, maceration	Mice	Extract: 500 mg/kg, i.p Isolated compounds 50, 100 mg/kg, i.p	The mixture of the isolated compounds synergized with each other.	[45]
	Whole plant	Methanol, maceration	High-fat diet followed by injection of streptozocin diabetic rats	Extract or solid lipid nanoparticles: 250 mg/kg, 4 weeks	The SLN preparation with the highest lipid content gave the best result of reduction of hyperglycemia and insulin resistance	[45]
*Retama raetam*	Fruit	Methanol, maceration	Streptozocin diabetic rats	100, 250, 500 mg/kg, 4 weeks	The extract significantly inhibits glucose absorption by rat isolated intestine	[48]
*Rhizophora mucronata*	Leaf	Water	Streptozocin diabetic rats	400 mg/kg, 6 weeks	Significant increase in the muscle levels of CAT, GSH and SOD	[31]
*Salvadora persica*	Root	Water, soxhlet	Streptozocin diabetic rats	250, 500 mg/kg, 21 days	Significant decrease in TC, TG, LDL, VLDL, increase in HDL	[49]
*Azadirachta indica*	Leaf	Water, percolation	Alloxan diabetic rabbits	200, 400 mg/kg, 25 days	Decrease in serum TG, cholesterol, LDL, increase in HDL	[51]
*Boswellia carterii*	Resin	Water, decoction	Streptozocin/Nicotinamide-diabetic rats	100 mg/kg, 4 weeks	Decrease in the serum cholesterol and TG	[58]
*Cissus rotundifolia*	Aerial parts	Water, maceration	Streptozocin/Nicotinamide-diabetic rats	100 mg/kg, 4 weeks	Decrease in serum cholesterol and TG	[58]
	Leaf	Water, maceration	Streptozocin/Nicotinamide-diabetic rats	100 mg/kg, 4 weeks	Significant decrease in urea, ALT, AST	[62]
*Dracaena cinnabari*	Resin	Ethanol 99%, maceration	Alloxan diabetic rats	100, 300 mg/kg, 14 days	Antihyperlipidemic effect	[64]
*Dracaena cinnabari*	Resin	Serial exhaustive extraction, soxhlet	In vitro MCF-7 cell line	100 μg/mL	Glucose uptake inducing activity of ethylacetate extract was found to be higher than Metformin	[63]
*Opuntia ficus-indica*	Seed	Hexane, p-ether, chloroform, maceration	Streptozocin-diabetic rats	0.2, 0.4, 0.6 g/kg, twice daily, 21 days	Reducing the expression of the PCK1 gene while increasing the expression of Slc2a2 gene in the liver tissue.	[66]
*Pulicaria inuloides*	Leaf	Oil extract, stem distillation	Streptozocin diabetic rats	400 mg/kg, 21 days	Inhibitory effect on *α*-glycosidase and *α*-amylase	[67]
*Solenostemma argel*	Leaf	Methanol, maceration	Methylprednisol-one diabetic rats	1 gm/kg, 14 days	Antioxidant activity	[69]

**Table 4 molecules-26-00742-t004:** Medicinal plants of Egypt ethnobotany.

Scientific Name	Common Name	Family	Traditional Use	Part Used	Reference
*Cassia acutifolia*	Not mentioned	Fabaceae	Constipation	Not mentioned	[141]
*Centaurea alexanderina*	Not mentioned	Asteraceae	Antimalarial, bitter tonic, diuretic, mild astringent, stomachic	Not mentioned	[130]

*Fraxinus ornus*	HabElderdar	Oleaceae	Diabetes	Not mentioned	[129]
*Moringa peregrina*	Hadendowa	Moringaceae	Edible, fever, headache, earache, burns, disinfectant, adenopathy, skin disorders, itching, soothe rash, purify water, wound, cancer, laxative, cathartic, malnutrients, ascites, leprosy, swellings	Flower, leaf, root, seed	[142,143]
*Nepeta Cataria*	Qatram, hashishat al-her (Arabic name)	Lamiaceae	Asthma, bronchitis, cough, diarrhea, gastrointestinal, respiratory hyperactivedisorders	Leaf	[135,144]
*Phoneix dactylifera*	Not mentioned	Arecaceae	Aphrodiasiac, tonic	Pollen, male flower	[145]
*Securigera securidaca*	Not mentioned	Fabaceae	Antidiabetic, antihyperlipidemic	Seed	[146]
*Trigonella stellate*	Not mentioned	Fabaceae	Dantidiabetic, antihyperlipidemic	Not mentioned	[137]
*Urtica pilulifera*	Nettle (Palestine)	Urticaceae	A diuretic, antiasthmatic, anti-inflammatory, hypoglycemic, hemostatic, antidandruff and astringent	Whole plant	[138,147]
*Zizyphus spina-christi*	Nabka, Seder (Palestine)	Rhamnaceae	Swellings, pain, and heat, eye inflammation, constipation, heartburn, diarrhea, wound, diuretic, liver problems, anus problems	Leaf, root, seed	[148]

**Table 5 molecules-26-00742-t005:** Medicinal Plants of Egypt reported constituents, use.

Scientific Name	Phytochemical Constituent	Pharmacological Use	Reference
*Cassia acutifolia*	Anthraquinone, phenolic glycoside	Laxative	[149,150]
*Centaurea alexanderina*	Sesquiterpene lactones, flavonoids	Hypoglycemia, cytotoxicity	[130]
*Cyperus laevigatus*	Quinones, flavonoids, sesquiterpenes, Steroids, essential oils	Anti-inflammatory, hepatoprotective, gastroprotective, antimalarial, antidiabetic activities	[131]
*Fraxinus ornus*	Alkaloid, coumarins, flavonoid, phenylethanoids, secoiridoid, tannin	Antimicrobial, antioxidant, anti-inflammatory	[129]
*Moringa peregrina*	Flavonoid	Antioxidant	[151]
*Nepeta cataria*	Essential oil, urosolic acid, *β*-sisoterol, campesterol, *α*-amyrin, *β*-amyrin, neptalactones, alkaloids, carenolides, tannins, saponins, coumarins	Antibacterial, antifungal, analgesic	[135,152]
*Phoneix dactylifera*	Phenolic acid, sterol, carotenoid, flavonoid, procyanidins, anthocyanins	Immune system, nephroprotective, hepatoprotective, gastrointestinal protective, antioxidant, antihyperlipidemic activity, anticancer	[145]
*Securigera securidaca*	Alkaloid, cardiac glycosides, coumarins flavonoid, saponins, tannins	Antihyperlipidemic, chronotropic, diabetes, diuretic, gastroprotective	[153]
*Trigonella stellate*	Flavonoids; isoflavonoid, phenolic compounds	Antihyperlipidemic, antioxidant	[137]
*Urtica pilulifera*	Lectins, b-sitosterol, phenolic acids triterpene	Antitumor, astringent, asthma, antidandruff, diuretic, diabetes, deputative, galactogogue	[154]
*Zizyphus spina-christi*	Triterpenoid saponin glycosides, polyphenol, tannin, flavonoid, cyclopeptides, cardiac glycoside, essential oil, alkaloid	Antinociceptive, antioxidant, antifungal, antibacterial, antidiabetic	[139,140]

**Table 6 molecules-26-00742-t006:** Medicinal plants with potential antidiabetic activity in Egypt.

Scientific Name	Part Used	Extraction Method, Solvent	Target	Intervention and Duration	Observations	Ref.
*Cassia acutifolia*	Leaf	Ethanol 80%, maceration	Nicotinamide-Streptozocin diabetic mice	10, 50 mg/kg, 1 week	No effect on serum insulin level	[129]
*Crataegus aronia*	Whole plant	Water, maceration	High-fat diet with small dose of streptozocin diabetic rats	500 mg/kg, 60 days	Lowered serum lipid levels, hepatic glycogen, hepatic lipid peroxidation, tumor necrosis factor *α*, interleukin, enhanced the level of reduced glutathione, superoxide dismutase, hepatic mRNA expression of the insulin receptor A isoform, glucose 6-phosphatase	[155]
*Centaurea alexanderina*	Leaf	Methanol 80%, reflux	Streptozocin diabetic rats	600 mg/kg, 60 days	Anti-inflammatory activity was seen for the extract	[130]
*Citrullus colocynthis*	Seed	Ethanol	Alloxan diabetic rats	50 mg/kg, 8 weeks	Decrease in lipid peroxidation, total cholesterol, triglyceride, total, direct bilirubin, increase on glutathione, lactate dehydrogenase, ALT, AST, ALP, total lipid were significantly increased in both normoglycemic and hyperglycemic rats	[156]
*Cyperus laevigatus*	Aerial parts	Methanol 70%, maceration	Streptozocin diabetic rats	50mg/kg, 14 days	Decreasing levels of NO, glucagon, promoted paraoxonase activity	[131]
*Fraxinus ornus*	Fruit	Ethanol 80%, maceration	Nicotinamide-Streptozocin diabetic mice	10, 50 mg/kg, 1 week	No effect on insulin serum level	[129]
*Moringa oleifera*	Leaf	Ethanol, maceration	Alloxan diabetic rats	150 mg/kg, 21 days	Quercetin has the greatest potential antidiabetic activity in the extract, followed by chlorogenic acid and moringinine	[157]
*Moringa peregrina*	Aerial parts	Ethanol 95%, percolation	Streptozocin diabetic rats	Extracts: 25 mg/kg, fractions: 50 mg/kg, 1 day	The *n*-hexane fraction was the only fraction that showed a highly significant antihyperglycemic activity	[18]
	Seed	ethanol 70%, soxhlet	Streptozocin diabetic rats	150 mg/kg, 30 days	The extracts exerted protective effects against lipid peroxidation ethanolic extract > aqueous extract	[19]
*Nepeta cataria*	Flowering aerial parts, seed	P-ether, ethyl acetate, ethanol 70%, soxhlet	Streptozocin diabetic rats	50 g/kg, 30 days	All extracts had significant scavenging of free radicals abilities, normalized liver function and inhibited lipid synthesis	[135]
*Phoenix dactylifera*	Seed	Aqueous suspension	Streptozocin diabetic rats	1 gm/kg, 4 weeks	Protective effects for kidney, liver	[133]
	Epicarp	Isolated compounds	Alloxan diabetic rats	20 mg/kg, 30 days	Significant decrease in ALT, AST, cholesterol, TG, increase in glutathione peroxidase and superoxide dismutase in liver	[158]
*Securigera securidaca*	Flower	Ethanol 90%, maceration	Alloxan diabetic rats	100, 200, 400 mg/kg	The extract was safe up to a dose of 2000 mg/kg (body weight), reduction in serum TG and total cholesterol levels	[136]
*Trigonella stellate*	Aerial parts, root	Ethanol 90%, maceration	Human hepatoma (HepG2) cell line	12.5, 25, 50 μg/mL	Activation of PPAR*α* and PPAR*γ* (fractions and isolated comp)	[137]
*Urtica pilulifera*	Aerial parts	Methanol, maceration	High-fat, low-dose streptozocin diabetic rats	250, 500 mg/kg	A significant antioxidant, anti-inflammatory effect	[138]

**Table 7 molecules-26-00742-t007:** Medicinal plants of Iran ethnobotany.

Scientific Name	Common Name	Family	Traditional Use	Part Used	Reference
*Allium paradoxum*	Alezi	Liliaceae	Acne, food flavoring	Raw vegetable	[161,163]
*Allium ascalonicum*	Not mentioned	Alliaceae	Diabetes	Not mentioned	[166]
*Allium sativum*	Sarimsaq, sir	Alliaceae	Diabetes, digestive system, blood fat, rabies	Bulb	[166,167,168]
*Arctium lappa*	Palvarg	Asteraceae	Antiscorbutic, antioxidant, blood purifiers, constipation, diuretic, disinfectant, gout	Berries, leaf, root	[169]
*Buxus hyrcana*	Shemshad	Buxaceae	Broken bone, toothache	Leaf	[167]
*Calendula officinalis*	Marigold, HamisheBahar	Asteraceae	Blood cleanser, eczema, dermal disorders, sudorific	Flower	[168,170]
*Camellia sinensis*	Chai Sabz	Theaceae	Anticancer, blood pressure, antihyperlipidemia, hepatitis, obesity	Leaf	[168,170]
*Convolvulus persicus*	Not mentioned	Convolvulaceae	Not mentioned	Not mentioned	[72]
*Cinnamomum zeylanicum*	Darchin	Lauraceae	Antibacterial, antioxidant, hypertension, diabetes	Bark	[164]
*Coriandrum sativum*	Geshniz	Apiaceae	Acne, antiseptic, appetizer, aphrodisiac, aromatic, calmative, flatulence, jaundice	Fruit	[168,170]
*Crataegus oxyacantha*	Sorkhevalik	Rosaceae	Diabetes, hypertension	Leaf	[164]
*Hibiscus sabdariffa*	Chay-e-makki	Malvaceae	Antioxidant, diabetes, hypertension	Flower	[164]
*Juglans regia*	Gerdu	Juglandaceae	Antidiarrheal, eczema, hair color	Fruit, leaf	[168]
*Morus alba*	Toot	Moraceae	Constipation, diabetes blood lipid reduction	Leaf	[164]
*Olea europaea*	Zeytoun	Oleaceae	Anti-hemorrhoids, blood lipid, diabetes, dermal allergy, hypertension, kidney stone, laxative, renal problems	Fruit, leaf, seed	[167,168,171,172]
*Parrotia persica*	Not mentioned	Hamamelidaceae	Antifever, broken bone, food coloring and flavoring	Bark	[163,173]
*Pimpinella affinis*	Taretizakebaghi	Apiaceae	Asthma, antimicrobial, antispasmodic, carminative, cholera, diuretic, migraine, narcotic	Flowering shoot, Seed	[163,174]
*Primula heterochroma*	Not mentioned	Primulaceae	Food flavoring	Not mentioned	[163]
*Portulaca oleracea*	Khorfeh	Portulaceae	Antibacterial, antiviral, diabetes, enhancing immunity	Seed	[164]
*Rubus fruticosus*	Tameshk	Rosaceae	Anti-infection, anticramp, hypertension, food flavoring, narcotic	Leaf	[164]
*Ruscus hyrcanus*	Butcher’s broom	Asparagaceae	Appetizer, antibleeding, antilaxative, anti-nephritis, anti-infection, antivaricose, aperient, diuretic, jaundice, laxative, vasoconstrictor	Leaves, fruit	[163,175]
*Smilax excelsa*	Not mentioned	Smilacaceae	Antieczema, diuretic, food flavoring	Not mentioned	[163]
*Syzygium aromaticum*	Alezi	Liliaceae	Acne, digestive disorder food flavoring	Raw vegetable	[164]
*Teucrium polium*	Kalpoore	Buxaceae	Analgesic, aperient, antiepileptic, antihairloss, antiheadache, anti-infection, antimalaria, antipneumonia, antirheumatism, carminative, constipation, febrifuge, stomachache	Aerial part	[164,176]
*Trigonella foenum-graecum*	Sic lefuit fenugreek	Fabaceae	Not mentioned	Not mentioned	[164,170]
*Urtica dioica*	Nettle	Urticaceae	Diabetes	Aerial parts, seed	[177]
*Urtica pilulifera*	Kara Isırgan (Turkey)	Urticaceae	Antidandruff, anti-inflammatory, asthma, astringent, blood purifier, diuretic, diabetes, enhancement of hemoglobin concentration, hemostatic, lower urinary tract infection, stimulating tonic (Egypt)	Whole plant, leaf	[138,178]
*Vaccinium arctostaphylos*	Darchin	Lauraceae	Antibacterial, antioxidant, blood pressure, diabetes	Bark	[164]

**Table 8 molecules-26-00742-t008:** Ethnobotany of plants form Iran.

Scientific Name	Common Name	Family	Traditional Use	Part Used	Reference
*Allium ampeloprasum*	Tarreh (Iran) Kurrat (Eygept)	Liliaceae	Asthma, antiseptic, diuretic, goat, vasodilatory, expectorant, constipation, hemoptysis, obesity, hemorrhoids, headache and as a diuretic, emmenagoug	Leaf, bulb	[182,183,277,278]
*Allium sativum*	Som, sir	Liliaceae	Antifungal, antiprotozoal, anthelminthic, antiviral, disinfectant, antitumor, gastric hepatic disorders, diabetes, hypertension, hypercholesterolemia, immunodeficiency syndromes	Bulb	[279,280,281]
*Amygdalus lycioides*	BadamTalkhkuhi	Rosacese	Antimicrobial, anti-inflammatory, diabetes	Aerial parts, root	[188]
*Amygdalus scoparia*	Badam-Koohi-Arzhan	Rosaceae	Appetizer, cardiovascular and respiratory diseases, headache, rheumatism, wound healing	Leaf	[281]
*Arctium lappa*	Baba adam	Asteraceae	Diabetes	Root, leaf	[190]
*Berberis integerrima*	Zarch	Berberidaceae	Abdominal ache, depurative, diabetes, hypertension	Fruit, root, leaf, stem	[176,282,283]
*Brassica napus*	Colza	Brassicaceae	anti-goat, anti-inflammatory, anti-scurvy, diuretic, bladder and hepatic and kidney colic	Root, seed	[284]
*Brassica rapa*	Kolza	Brassicaceae	Diabetes	Root, seed, leaf	[285]
*Capparis spinosa*	Shomisheytoni	Capparaceae	Diabetes, diuretic, gout arthritis, liver disorders, neurological conditions, rheumatoid, paralysis	Fruit, leaf	[283,286]
*Centaurea bruguierana*	Baad-Avard	Asteraceae	Hypoglycemic	Aerial fruiting parts	[201]
*Cichorium intybus*	Chicory or Kasni	Asteraceae	Antipyretic, depurative, diabetes, choleretic, eupeptic, hypotension, laxative, stomachic, tonic	Whole plant	[202,203]
*Citrullus colocynthis*	Hanzal	Cucurbitaceae	Abortifacient, antiepileptic, analgesic, anti-inflammatory, diabetes, hair growth-promoting	Seeds, root, pulp, fruits	[279,287]
*Cornus mas*	Not mentioned	Cornaceae	Anemia, chickenpox, diarrhea, diabetes, digestion problems, hepatitis A, heal wounds inflammatory bowel disease, fever, kidney stones, malaria, measles, pyelonephritis, rickets, sore throat, sunstroke, urinary tract infections, ulcer	Leaf, flowers, fruits	[208,288]
*Cucumis sativus*	Tokhm-e-khiyar	Cucurbitaceae	Anti-fever, demulcent, diabetes	Seed	[212]
*Cucurbita pepo*	Kadohalvai	Cucurbitaceae	Diabetes	Seed, fruit	[285]
*Eryngium caucasicum*	Aweiyeh	Apiaceae	Flavoring	Leaf	[289]
*Eucalyptus globules*	Okaliptus	Myrtaceae	Diabetes	Leaf	[290]
*Falcaria vulgaris*	Paghazou	Apiaceae	Diabetes, gastric and duodenal ulcers, wound healing	Stem, leaf	[291]
*Ferula assafoetida*	“Anghouzeh”, “Khorakoma” “Anguzakoma”	Apiaceae	Asthma, epilepsy, flatulence, influenza, intestinal parasites, stomach ache, weak digestion	Oleo-gum resin	[221]
*Galega officinalis*	Galega	Fabaceae	Diabetes	Flower	[290]
*Gundelia tournefortii*	Kangar	Asteraceae	Antiparasite, diabetes	Leaf, root	[281,292]
*Heracleum persicum*	Glopar	Apiaceae	Carminative	Fruit	[293]
*Hordeum vulgare*	Jo dosar	Poaceae	Diabetes	Seed, bran	[285]
*Juglan regia*	Gerdoo	Juglandaceae	Antidiarrhea, antiparasitic, blood purifier, diabetes, hemorroides, venous insufficiency	Leaf	[228,281]
*Mentha spicata*	Nana, pooneh	Lamiaceae	Antispasmodic, carminative, diabetes, digestive, stomach pain-relieving agent,	Aerial parts, leaves, essence	[233]
*Nasturtium officinale*	Bakalu	Brassicaceae	Jaundice in children, abortion	Aerial parts, leaf	[292]
*Otostegia persica*	ShekarShafa	Lamiaceae	Antipyretic, antihyperlipidemia, analgesic, arthritis, cardiac distress, carminative, cough, diabetes, headache, laxative, parasite repellent, reducing palpitation, regulating blood pressure, rheumatoid toothache, stomachache, morphine withdrawal	Aerial Parts	[285,294]
*Phoenix dactylifera*	Khorma, Tarooneh	Arecaceae	Aphthous, antidepressant, bladder and nerve tonic, diabetes, pertussis, rheumatoid arthritis sedative, tranquilizer, wound healer	Spathe, pollen, leaf	[145,295,296,297]
*Pistacia atlantica*	Zhevi	Anacardiaceae	Diabetic ulcers, hyperglycemia	Juice	[283]
*Punica granatum*	Anar	Punicaceae	Astringent, burns, diarrhea, hematopoiesis, gastrointestinal disorders inflammation, pain, oral aphthous, sore throat	Fruit	[298,299]
*Pyrus boissieriana*	Not mentioned	Rosaceae	Anticramp, antihypertensive, food flavoring, anti-infection, narcotic,	Not mentioned	[163]
*Rheum turkestanicum*	Eshghan	Polygonaceae	Depurative, diabetes, hypertension jaundice	Root	[300]
*Rhus coriaria*	Sumac	Anacardiaceae	Adjustment blood lipid, diabetes, diarrhea, dysentery, hemorrhoids, gout, reduction of uric acid, wound healing table spice	Not mentioned	[248,249]
*Salvia hydrangea*	Gol-e Arooneh		Anti-inflammatory, antispasmodic, analgesic, cold	Leaf	[253]
*Salvia hypoleuca*		Lamiaceae	diabetes, hemorrhoids, insecticide, purgative		[301]
*Salvia officinalis*	Maryam	Lamiaceae	Antiseptic diabetes, diuretic, dyspepsia, emmenagogue, fever	Leaf, flower	[302]
*Securigera securidaca*	GandehTalkheh	Fabaceae	Diabetes, hypertension, hyperlipidemia	Seed	[256]
*Solanum nigrum*	Hava	Solonaceae	Antibacterial, anticonvulsant, antipyretic, antioxidant, anti-inflammatory, antitumorigenic, cytotoxicity, diabetes, diuretic, enteric diseases, fever, hepatoprotective, inflammation, mycotic infection, pain, ulcer, sexually transmitted diseases	Aerial parts, fruit	[257,285]
*Teucrium polium*	Kalpooreh	Lamiaceae	Abdominal pain, cold, diabetes, indigestion, urogenital diseases	Aerial parts	[259]
*Trigonella foenum-graecum*	Shanbalileh	Fabaceae	Backache, bladder cooling reflex, cold, cough, diabetes, emollient, hepatitis, gastrointestinal disorders, loss of appetite, mouth odor, splenomegaly, undesired odor of body and sweat, skin disease, red spot of eye, tonic	Leaf, seed, aerial part	[285,303]
*Urtica dioica*	Gazane, chitchitiodghin, Gazgazuk, Aragh	Urticaceae	Allergic rhinitis, diabetes, digestive, diuretic, hypertension, galactogogue, inflammation, prostatic hyperplasia, rheumatoid arthritis, tonic	Root, aerial parts, leaf	[268,284]
*Vaccinium arctostaphylos*	Qaraqat, Cyah-gileh	Ericaceae	Diabetes, hypertension	Berries	[265]
*Zataria multiflora*	Avishan	Lamiaceae	Anesthetic, antinociceptive, antispasmodic, flavoring-preserving foods/drinks, irritable bowel syndrome, respiratory tract infections	Not mentioned	[304]
*Zizyphus spina-christi*	Sedr	Rhamnaceae	Antiseptic, antifungal, anti-inflammatory, bronchitis, cough, dysentery, nausea, sedative, skin diseases, wound healing, tuberculosis, vomiting, abdominal pains associated with pregnancy	Leaf, fruit, seed	[305]

**Table 9 molecules-26-00742-t009:** Medicinal plants of Iran constituents, use.

Name	Chemical Constituents	Biological and Pharmacological Activities	Reference
*Allium ampeloprasum*	Cinnamic acid derivatives, nitrates, flavonoids, polysaccharides, glucosinolates, organosulphur compounds	Antibacterial, antioxidant, antifungal, protect skin against damage due to pathogenic agents, alleviate gastrointestinal diseases, inflammatory, hepatotoxicity	[183,277,306]
*Allium ascalonicum*	Volatile sulfur compounds, saponins, flavonoids, furostanol saponin	Antibacterial, anti-fungal, antihelicobacter pylori, beneficial hematological influences, antioxidant, peroxynitrite-scavenging capacity	[307,308]
*Allium sativum*	Organo-sulphur compounds, volatile sulphur compounds (diallyl sulphide), polyphenols	Hypoglycemic, cardiac diseases, antiseptic, toothache, antihyperlipidemia, anthelmintic, antihypertensive	[166,168,309]
*Amygdalus lycioides*	Alkaloids, amygdalin, flavonoid, phenolic compound, terpenoids,	Antioxidant, anticancer, anti-inflammatory, antiaging, antimicrobial, decrease the risk of colonic cancer, increase HDL cholesterol, reduce LDL cholesterol, cardioprotective	[189,310]
*Amygdalus scoparia*	Amygdalin, flavonoids, amino acids, linoleic acid	Antioxidant	[311]
*Arctium lappa*	Flavonoids, lignans, phenols, saponins, tannin	Anti-inflammatory, hepatoprotective, free radical scavenging activities	[190]
*Berberis integerrima*	Anthocyanins, alkaloids, berberine	Anticonvulsant, bleeding, diarrhea, fever, gastrointestinal disease, hepatitis, malaria, swollen gums teeth, sore throat, bile inflammation, reducing blood cholesterol	[191,312]
*Brassica napus*	Anthocyanin, cinnamic acid hydroxyl derivatives, flavonoids, erucic acid	Antioxidant	[194,284]
*Brassica rapa*	Flavonoids, phenylpropanoid derivatives, indole alkaloids, sterol glucosides, glucosinolates and isothiocyanate	Analgesic, anticancer, anti-inflammatory, antimicrobial, antioxidant, cardiovascular and hypolipidemic effect, diabetes, hepatoprotective, metabolic syndrome neuroprotective, obesity	[195,313]
*Capparis spinosa*	alkaloids, lipids, polyphenols, flavonoids, Tocopherols, carotenoids, glycosides, capparic acids, tannins	Antioxidant, antimicrobial, anticancer, antiallergic, hepatoprotective effects	[199,286]
*Centaurea bruguierana*	Sesquiterpene lactone, flavonoid (kaempferol, rutin, quercetin)	Antiplasmodial, antipeptic ulcer	[201]
*Cichorium intybus*	Aliphatic compounds, terpenoids, saccharides, methoxycoumarin cichorine, flavonoids, essential oils and anthocyanins	Antimicrobial, anthelmintic, antimalarial, diabetes, hepatoprotective, gastroprotective, anti-inflammatory, analgesic, antioxidant, tumor inhibitory, antiallergic	[202]
*Citrullus colocynthis*	a bitter compound (Cucurbitacins), colocynthein, colocynthetin, pectin, gum, flavonoids, alkaloid (choline), volatile terpenoids, fixed oil albuminoids, tocopherols and carotenes	Antidiabetic, hypolipidemic, antimicrobial, anti-inflammatory, antioxidant, cytotoxic, insecticidal, antiallergic	[206,287]
*Cornus mas*	Sugar, organic acids, tannins, anthocyanins, phenolic acid, tannin, vitamin C, flavonoid, iridoids, terpene (mono, tri), Carotenoids	Antimicrobial, antidiabetic, antiobesity, hypolipidemic and anti-atherosclerotic, cytotoxicity, hepatoprotective, renalprotective, neuroprotective, anti-inflammatory, antioxidant, antiplatelet, cardioprotective, antiglaucomic, reproductive organ -protective, radioprotective, aldose redustase inhibitory	[231,288,314]
*Cucurbita pepo*	Tetra cyclic triterpens, saponins, proteins, fibers, polysaccharides, miberal, carotenoid, tocopherol	Antioxidant, lipid-lowering, hepatoprotective, anticarcinogenic, antimicrobial, antidiabetic	[213]
*Cucumis sativus*	Flavonoids, saponin, tannin, steroid	Antimicrobial, antitumor, antacid, carminative, wound healing, against ulcerative colitis, skin irritation, hypoglycemic and hypolipidemic	[315]
*Eucalyptus globulus*	Euglobals, essential oils, hydrocarbons	Antidiabetic, anti-bacterial, antiplaque, antitumor, antiviral, antifungal, antihistaminic, anti-inflammatory, antioxidant, antimalarial	[217]
*Falcaria vulgaris*	Volatile oil, phenolics, flavonoid,	Anti-inflammatory, antioxidant, antibacterial, antifungal, antiviral, and bleeding inhibitor activities	[316,317]
*Ferula assafoetida*	Coumarine, flavonoids, gum, phenolic acids terpene, volatile oil	Antioxidant, antiviral, antifungal, cancer chemopreventive, antidiabetic, antispasmodic, hypotensive, molluscicide	[220,221]
*Galega officinalis*	Alkaloid, flavonoid	Antioxidant	[318]
*Gundelia tournefortii*	Coumarin, terpene, sterol, essential oil, phenolic compounds,	Antibacterial, anti-inflammatory, hypolipemic activity, hepatoprotective, antiplatelet, antioxidant	[319]
*Heracleum persicum*	Alkaloids, flavonoids, furanocoumarin, terpenoids; triterpenes, volatile substances	Anti-inflammatory, immunomodulatory, growth enhancer, antioxidant, anticonvulsant, analgesic, hypocholesterolaemic agent	[320,321]
*Hordeum vulgare*	Soluable fiber components especially *β*-glucans	Anti-inflammatory, antilactagogue, antimutagenic, antiviral, astringent, antioxidant, antiprotozoal, aphrodisiac, demulcent, digestive, diuretic, expectorant, febrifuge, hypocholesterolemic, refrigerant, sedative, stomachic, tonic, poultice for burns and wounds	[322]
*Juglans regia*	Flavonoids, phenolics, tannins, flavonoids, phytosterols, tocopherol	Antioxidant, antidiabetic, antihypertensive, antimicrobial, anticancer, liver and kidney protection, lipid-lowering effect	[228]
*Mentha spicata*	Flavonoids, terpenoids, monoterpenes phenols	Antimicrobial, antispasmodic, antiplatelet, insecticidal, neutraceutical and cosmetic industries	[233]
*Nasturtium officinale*	carotenoids, polyphenols, vitamin C, vitamin A and *α*-tocopherol	Antimicrobial, antioxidant, antiestrogenic, anticarcinogenic, for the prevention of cancer	[234]
*Otostegia persica*	Alkaloids, essential oil, flavonoids, tannin, terpenes, mono, di and sesquiterpene, steroids	Antimicrobial, antioxidant, antidiabetic, antiglycation, anti-aphids, hepatoprotective	[235,294]
*Olea europaea*	Lignan like compounds, lignan glycoside, coumarin, flavonoid, oleuropein, phenolics, hydroxytyrosol derivatives, secoiridoid, secoiridoid glycosides, triterpene	Antidiabetic, anticancer, antimicrobial, antioxidant, antihypertensive, enzyme inhibitory activities, anti-inflammatory, antinociceptive activities, gastroprotective, neuroprotective	[323]
*Phoenix dactylifera*	Alkaloids, anthocyanins, carotenoids, flavonoids, phenolics, procyanidins, sterols, vitamins, tannins.	Anticancer, antioxidant, antimutagenic, antihemolytic, antiviral, antifungal, anti-inflammatory, hepatoprotective, gonadotropic activity immunostimulant, nephroprotective,	[145]
*Punica granatum*	Alkaloid, anthocyanin, catechin, ellagitannins, flavonoid, sterol, tannins	anticancer, antidiabetic, anti-inflammatory, antimicrobial, healing activity	[324]
*Pyrus boissieriana*	Arbutin, glycosides, flavonoids, phenols	Antioxidant, antihyperlipidemic, bactericidal and antifungal effects, diabetes	[163,240]
*Rheum turkestanicum*	Anthraquinone, flavonoid, phytosterol, phenolic glycosides	Anticancer, cardioprotective, diabetes, nephroprotective, hepatoprotective, neuroprotective	[250]
*Rhus coriaria*	tannins, anthocyanins, various organic acids such as malic and citric acids, fatty acids, vitamins, flavonoids and terpenoids, essential oil	Antimicrobial, antifungal, antiviral, antioxidant, anti-inflammatory, hepatoprotective, xanthine oxidase inhibition, hypoglycemic, cardiovascular protective activities	[248]
*Salvia hydrangea*	Flavonoids, phenolic acid, terpenoids	Antioxidant	[253]
*Salvia hypoleuca*	Essential oil, lactones, isomeric epoxides, monolactone and hypoleuenoic acid, sterols, terpenes, sesqui, di and triterpenes	Antioxidant	[325,326]
*Salvia officinalis*	Flavonoids, carnosic acid, rosmarinic acid	Antioxidant	[302]
*Securigera securidaca*	Cardenolides, coumarins, dihydrobenzofuran derivatives, flavonoids, steroidal and pentacyclic triterpenoid-type saponins	Antiulcerogenic, antiepileptic, chronotropic, diuretic, hypokalemic	[136,256]
*Solanum nigrum*	Catechin, caffeic acid, epicatechin, flavonoids, glycoalkaloids, glycoproteins, polysaccharides, polyphenolic compounds, protocatechuic acid	Antimicrobial, anti-HCV, anti-ulcer, analgesic, anti-inflammatory, antidiarrheal, anticancer, antiseizure, cardioprotective, cytotoxic activity diabetes, immunostimulant, hepatoprotective, larvicidal activity	[257]
*Teucrium polium*	Terpenoids, mono, di and sesquiterpene, flavonoids, neoclerodane, polyphenols	Antioxidant, antimutagenic, anticancer, anti-inflammatory, antinociceptive, antispasmodic, antiulcer, antimicrobial, diabetes, hepatoprotective, hypolipidemic	[258,259]
*Trigonella foenum-graecum*	Flavonoids, alkaloids, coumarins, vitamins, steroidal saponins	Anti-inflammatory, antilipidemic, antioxidant, antimicrobial, antiulcer, anticarcinogenic, carminative, diabetes, hypocholesterolemic, hepatoprotective, galactogogue	[261,285,327]
*Vaccinium arctostaphylos*	Anthocyanins, flavonoids, phenolic acids	Antioxidant, antimicrobial	[265,328]
*Urtica dioica*	Alkaloids, agglutinin, lecithin, flavonoid, phenolic acids, stigmasterol, terpenes, coproporphyrin, lignan, and violaxanthin, coxamarins	Antiproliferative, antioxidant, antidandruff, anti-inflammatory, antimicrobial, cancer, coronary heart disease, diabetes, joint pain reduction, urinary tract infection, psychotic disorders, viral and parasitic diseases	[271]
*Zataria multiflora*	Volatile oil	Antioxidant, antinociceptive, anti-inflammatory, antibacterial, antiviral, antifungal, immunostimulant, pain-relieving	[273,329]
*Zizyphus spina-christi*	Flavonoid, isoquinoline alkaloid, cyclopeptide, saponin, triterpenes	Analgesic effect, antioxidant, antidiabetic, antifungal, antibacterial, antinociceptive	[275,305]

**Table 10 molecules-26-00742-t010:** Plants with antidiabetic potential from Iran.

Scientific Name	Part Used	Extraction Method, Solvent	Target	Intervention and Duration	Observations	Ref.
*Allium ampeloprasum*	Bulb	Ethanol 70%, maceration	Alloxan diabetic rats	400, 800 mg/kg, 8 weeks	Prevent diabetes associated complications; decreased the levels of serum lipids and MDA, significantly increased the activity of antioxidant enzymes	[183]
*Allium ascalonicum*	Bulb	Methanol 80%, soxhlet	Alloxan diabetic rats	250, 500 mg/kg, 3 weeks	Elevated expression of both insulin and glucose transporters transcripts	[166]
*Allium sativum*	Bulb	Methanol 80%, soxhlet	Alloxan diabetic rats	250, 500 mg/kg, 3 weeks	Elevated expression of both insulin and glucose transporters transcripts	[166]
*Arctium lappa*	Root	Methanol 40%, maceration	nicotinamide-Streptozocin diabetic rats	200, 300 mg/kg, 28 days	Decreased level of triglyceride, vLDL, and alkaline phosphatase	[190]
*Amygdalus lycioides*	Spach branches	Methanol 50%, maceration	Streptozocin diabetic rat	125, 250, 500, 1000 mg/kg, 2 weeks	Decreased total cholesterol, LDL, TG, Cr, and alkaline phosphatase levels, increased ALT, ASTT, total number and numerical density of *β*-cells increased	[188]
*Amygdalus scoparia*	Fruit	Hexane, maceration	Streptozocin diabetic rat	1 mL/kg 15 days	Improve dregeneration of B cells	[189]
	Leaf	Aqueous, hydroalcoholic solvent	Streptozocin diabetic rats	30, 60, 120 mg/kg, 3 days	Increase in the serum insulin level	[330]
*Avicennia marina*	Leaf	Water, soxhlet	Streptozocin diabetic rats	100 and 200 mg/kg, i.p., one month, alternate day	Decreased the serum levels of the liver enzymes and tissue level of MDA, and the activity of the liver tissue’s antioxidant enzymes was increased	[32]
*Berberis integerrima*	Root	Water, maceration	Streptozocin diabetic rat	250 and 500 mg/kg, 6 weeks	Significant decrease in TG, cholesterol, LDL, ALT, AST, ALP, total bilirubin, Cr and urea, increase in HDL-cholesterol and total protein	[191]
	Fruit	Water, maceration	Streptozocin diabetic rats	250 and 500 mg/kg, 6 weeks	Extract did not possess the hypoglycemic and hypolipidemic activity.	[192]
	Fruit	Anthocyanin fraction	Streptozocin diabetic rats	200, 400 and 1000 mg/kg	Significantly increased liver glycogen and body weight	[193]
*Brassica napus*	Juice	Water, decoction	Alloxan diabetic rats	4 weeks	Decreased TG, cholesterol and LDL	[194]
*Brassica rapa*	Leaf	Water, maceration	Alloxan diabetic rats	200, 400 mg/kg	Decreased ALT, cholesterol, LDL, HDL, increased TG, AST	[195]
*Capparis spinosa*	Fruit	Ethanol 80%, soxhlet	Alloxan diabetic rats	300 mg/kg, 12 days	There was no evidence of regeneration in the liver of diabetic rats, brings the blood glucose significantly toward normal values from day 2 onwards	[198]
	Fruit	Ethanol 70%, maceration, soxhlet	Streptozocin diabetic rats	5 mg/kg	Dose-dependent decrease in blood sugar, decrease in triglycerides	[197]
	Root	Ethanol 70%, maceration	Streptozocin diabetic rats	0.2, 0.4 g/kg	Decreased LDL, AL, ALP, insulin levels did not increase, increased HDL	[331]
	Fruit	Water, decoction	Streptozocin diabetic rats	20 mg/kg, 28 days	No significant influence on the insulin level, decreased blood TG, cholesterol content, reduced the mRNA expression, enzyme activities of glucose-6- phosphatase and phosphoenolpyruvate carboxykinase in liver	[200]
*Centaurea bruguierana*	Aerial fruiting parts	Water, dichloromethane, ethylacetate, methanol, percolation	Streptozocin, alloxan diabetic rats	200, 400 mg/kg	Aqueous extract showed best effect	[201]
*Cichorium intybus*	Seed	Water, maceration	Streptozocin, Streptozocin and niacinamide diabetic rats	-------	Normalization of blood ALT, TG, total cholesterol, HB1C	[203]
*Citrullus colocynthis*	Seed	Ethanol 80%, maceration	Alloxan diabetic rat	300 mg/kg, 12 days	Enhanced regeneration of B-cells, increased size of pancreatic islet, improvement of hepatic tissue	[207]
*Cornus mas*	Fruit	Ethanol 70%, maceration	Alloxan diabetic rat	2 gm/kg, 1 month	Less severe hepatic portal inflammation, antidyslipidemia effects	[208]
*Cucurbita pepo*	Fruit	Powder	Alloxan diabetic rat	1, 2 gm/kg 4 weeks	Reduced cholesterol, TG, LDL and CRP levels	[213]
*Cucumis sativus*	Seed	Ethanol 75%, percolation buthanol, maceration	Normal and streptozocin diabetic rat	0.2, 0.4, 0.8 g/kg, 9 days	The extracts were not effective in reducing blood glucose levels in normal and diabetic rats	[212]
*Eucalyptus globulus*	Leaf	Water, decoction	Streptozocin diabetic mice	20, 62.5 g/kg eucalyptus in the diet, and 2.5 g/L extract in drinking water, 4 weeks	Dose-dependent amelioration of diabetic states by partial restoration of pancreatic *β*-cells	[216]
	Leaf	Water, hot maceration	Streptozocin diabetic rats	2.5 mg/mL, 4 weeks	Increase ALT, AST and ALP activity	[332]
*Falcaria vulgaris*	Aerial parts	Water, maceration	Streptozocin diabetic rat	200, 600, 1800 μg/kg	Hematoprotective and nephroprotective activity	[218]
*Ferula assa-foetida*	Oleo-gum resin	Ethanol 70%, maceration	Streptozocin diabetic rat	150, 250 mg/kg, 6 weeks	Protective effect of liver and kidney damage	[221]
	Oleo-gum resin	Water, maceration	Streptozocin diabetic rat	50, 100, 300 mg/kg, 4weeks	50 mg/kg significantly lowered the serum glucose concentration	[219]
*Galega officinalis*	Leaf	Powder	Streptozocin diabetic rats	1.5, 3 g/kg	Body weight-reducing properties	[223]
	Root	Ethanol, maceration	Streptozocin diabetic rats	150 mg/kg, o.p, i.p, 6 weeks	Decreased LDL, leptin, increased TG, VLDL, AST	[333]
	Not mentioned	Hydroalcoholic solvent	Streptozocin diabetic rats	50 mg/kg, i.p, 20 days	Decreased urea and creatinine	[224]
*Gundelia tournefortii*	Aerial parts	Water, maceration	Alloxan diabetic rats	5, 10, 20, 40 mg/kg, 20 days	Alleviation of diabetic complications such as nephropathy	[225]
	Shoots	Water	Streptozocin diabetic rats	400 mg/kg, 21 days	Recovery of pancreas tissue	[226]
*Horddeum vulgare*	Seed	Ethanol 75%, percolation and alkaline extract	Streptozocin diabetic rats	0.1, 0.25, 0.5 g/kg, 11 days	Restored body weight, long term benefits	[227]
	Not mentioned	Hexane oil extraction	Alloxan diabetic rats	i.p administration of oil, 6 weeks	Reduced insulin, HbA1C, total cholesterol, LDL, VLDL, HDL and TG	[229]
*Juglans regia*	Leaf	Ethanol 90%, maceration	Streptozocin-nicotinamide diabetic rats	200 mg/kg, 1 month	Reduced HbA1C, total cholesterol, LDL and TG	[230]
	Leaf	Methanol 70%, maceration	Streptozocin diabetic rats	200, 400 mg/kg, 4 weeks	Significant increase in diameter and number of *β*-cells compared to diabetic control group	[231]
	Leaf	Methanol 70%, maceration	Alloxan diabetic rats	250, 500 mg/kg, 3 weeks	Significant inhibition of *α*-glucosidase, maltaseand sucrase enzymes	[232]
*Mentha spicata*	Leaf	Water, soxhlet	Alloxan diabetic rats	300 mg/kg, 21 days	LD_50_ ˃ 1500 mg/kg (body weight), decreasedcholesterol	[233]
*Nasturtium officinale*	Aerial parts	Water, maceration	Streptozocin diabetic rats	100, 200 mg/kg, 4 weeks	Decrease in total cholesterol and LDL	[234]
*Otostegia persica*	Aerial parts	Water, decoction	Streptozocin diabetic rats	100, 200, 400 mg/kg, 1 month	Reduced TG	[235]
	Aerial parts	Methanol, soxhlet	Streptozocin diabetic rats, C187 *β*-cell line	200, 300, 400 mg/kg	Decreased MDA and increased GSH levels in the liver	[236]
*Otostegia persica*	Aerial parts	Ethanol 50%, percolation	Streptozocin diabetic rat	500 mg/kg	Helped prevent the entering of the remaining *β*-cells into some pathologic changes, such as hypertrophy	[237]
*Phoenix dactylifera*	Leaf	Ethanol extract, maceration	Alloxan diabetic rat	Extract: 100, 200, 400 mg/kg, 14 days Fractions: 50, 100, 200 mg/kg, 14 days	Decrease in water intake, serum TG and cholesterol, increase in plasma insulin level	[246]
*Pistacia atlantica*	Fruit	Hexane extract, maceration	Streptozocin diabetic rat	1 mL/kg, 15 days	Improvedregeneration of *β* cells	[189]
*Punica granatum*	Fruit	Methanol extract, maceration	Streptozocin diabetic guinea pigs	500 mg/kg, 4 weeks	Enhanced glutathione content as well as the activity of catalase, decrease in total cholesterol and TG	[247]
*Pyrus boissieriana*	Leaf	Methanol, maceration	Alloxan diabetic rat	500, 1000 mg/kg, 4 days	Decreased serum TG cholesterol, increased antioxidant status	[239,240]
*Rheum turkestanicum*	Rhizome	Water, decoction	Streptozocin diabetic rats	200, 400, 600 mg/kg, 3 weeks	Decreased triglycerides, no hypoglycemic or hepatoprotective effect in diabetic rats	[251]
*Rhus coriaria*	Fruit	Ethanol, maceration	Alloxan diabetic rats	200, 400 mg/kg, 21 days	Increased serum HDL, superoxide dismutase and catalase activities, reduced LDL, inhibited maltase and sucrase activities	[249]
	Not mentioned	Water	Alloxan diabetic rats	50, 100, 250 mg/kg, 28 days	Increase catalase activities in liver and kidney	[248]
*Salvia hydrangea*	Aerial parts	Ethanol 90%, maceration	Streptozocin diabetic rats	100, 200 mg/kg, 21 days	Reduce blood fat	[253]
*Salvia hypoleuca*	Whole plant	Ethanol, maceration	Alloxan and normal diabetic rat	250, 450 mg/kg, 14 days	Non-significant reductions for the non-diabetic groups that received extracts.	[252]
*Salvia officinalis*	Leaves	Methanol, soxhlet	Alloxan diabetic rat	250, 500 mg/kg, 21 days	Elevated expression of both Ins and Glut-4 transcripts	[166]
*Securigera securidaca*	Seed	Aqueous suspension	Alloxan diabetic rat	2, 4 g/kg, 3 days, intra-gastric gavage	Protective effect against oxidative stress	[254]
	Seed	CCl_4_, ethanol70%, dichloromethane, maceration	Streptozocin diabetic rat	200 mg/kg, 16 days	Carbon tetrachloride extract showed the best and most significant hypoglycemic and hypolipidemic activities with a *β*-cells-protecting effect from high glucose-induced apoptosis, and also increased insulin level and sensitivity	[255]
	Seed	Ethanol 70%, maceration	Streptozocin diabetic rat	100, 200 mg/kg, 4 weeks	Decreased serum total cholesterol, LDL, increased HDL	[334]
	Seed	Methanol 80%, chloroform fraction, maceration	Streptozocin diabetic rat	Methanol: 10, 400 mg/kg Fraction: 400, 600 mg/kg	Securigenin glycosides (isolated) reduced blood glucose equivalent to glibenclamide and elevated insulin level to normal	[256]
*Solanum nigrum*	Fruit	Water, decoction	Streptozocin diabetic rat	1 gm/L, 8 weeks	Improved lipid profile, decreased Ca/Mg ratio, decrease vessel atherosclerosis and prevented diabetic vesselcomplications	[306]
*Teucrium polium*	Leaf	Water, decoction	Streptozocin diabetic rats	100 mg/kg, 3 weeks	No decrease in body weight for diabetic rats	[259]
*Trigonella foenum-graecum*	Whole plant	Water, decoction	High-fructose diet diabetic rat	10% of extract 8 weeks	The extract improve insulin resistance	[264]
	Seed	Methanol, maceration	Streptozocin diabetic guinea pigs	500 mg/kg, 4 weeks	Enhanced GSH content as well as the activity of catalase, decrease in total cholesterol and TG	[247]
*Urtica dioica*	Whole plant	Aqueous distillate, distillation	Streptozocin diabetic rat	12.5 mL/kg 1 month	Prevents islet atrophy and/or regenerate pancreatic *β*-cells	[269]
	Leaves	Ethanol 70%, maceration	Fructose induced diabetes	50, 100, 200 mg/kg, 2 weeks	Decreased LDL, leptin, LDL/HDL ratio, FIRI (fasting insulin resistance index), increased serum TG, VLDL, and AST	[335]
*Urtica dioica*	Whole plant	Water, decoction	High fructose diet diabetic rat	10% extract, 8 weeks	Urine glucose decreased significantly	[264]
	Leaves	Water, infusion	Streptozocin diabetic rats, RIN-5F and L6 myotubes cell lines	6.25 mg/kg, 1.25 g/kg 1 month	Regeneration and less *β* cell damage, increased insulin secretion in the RIN-5F cells and glucose uptake in the L6 myotubes cells, lower TG and cholesterol	[271]
*Vaccinium arctostaphylos*	Fruit	Ethanol 95%, maceration	Alloxan diabetic rat	200, 400 mg/kg, 21 days	Decreased total cholesterol and TG	[266]
*Vitex agnus-castus*	Fruit	Ethanol 70%, maceration	D-galactose-induced aging mouse	500, 600 mg/kg, twice daily, 7 days	Pancreatic protective effects in natural aged and aging model mice	[267]
*Zataria multiflora*	Leaf	Hydrodistillation	Streptozocin diabetic rats	50 μL/kg 28 days	Decrease in in plasma ALP, AST, ALT, and significant increase in total protein and insulin	[273]
*Ziziphus vulgaris*	Fruit	Water, maceration	Onreptozocin diabetic rats	0.25, 0.5, 1, 1.5, 2 g/kg 14 days	Decrease in LDL and TG	[275]

**Table 11 molecules-26-00742-t011:** Medicinal plants of Iraq ethnobotany.

Scientific Name	Common Name	Family	Traditional Use	Part Used	Reference
*Bauhinia variegate*	Mountain ebony, orchid-tree, poor-man’s orchid, camel’s foot, Napoleon’s hat	Fabaceae	Anthelmintic, astringent, bronchitis, diabetes, diarrhea, laxative, leprosy, piles, tumors, tonic, worm infestations	Stem bark, leaves, buds	[18]
*Momordica charantia*	Kerela	Cucurbitaceae	Anthelmintic, hemmoroid, gout, stomachic	Fruit	[345]
*Rheum ribes*	Rawand, Rewas	Polygonaceae	Anthelmintic, diabetes, diarrhea, expectorant, hypertension, obesity, ulcer	Root	[346]

**Table 12 molecules-26-00742-t012:** Medicinal plants of Iraq constituents, use.

Scientific Name	Phytochemcal Constituent	Pharmacological Use	Refernce
*Bauhinia variegate*	Cardiac glycosides, flavonoids, reducing sugars, saponins, steroids terpenoids, tannins	Anticancer, antioxidant, antimicrobial, anti-inflammatory, antiulcer, hepatoprotective, hypolipidemic, immunomodulating, nephroprotective, molluscicidal and wound healing effects	[347]
*Momordica charantia*	Alkaloid, lipid, phenolics, triterpene, saponin, steroid	Diabetes	[339]
*Rheum ribes*	Anthraquinone, flavonoid, phenolic compounds, stilbene,	Antidiabetic, antioxidant	[341,348]

**Table 13 molecules-26-00742-t013:** Plants with antidiabetic potential form Iraq.

Scientific Name	Part Used	Extraction Method, Solvent	Target	Intervention and Duration	Observations	Ref
*Achillea santolina*	Leaf	Water, infusion	Streptozocin diabetic rats	150, 250 mg/kg, 28 days	Dose-dependent hypoglycemic response	[349]
*Bauhinia variegate*	Leaf	Ethanol 70%, maceration	Dexamethasone diabetic rats	200 mg/kg, 28 days	Decrease in TC and TG, increase in HDL	[338]
*Momordica charantia*	Seed	Water, maceration	Alloxan diabetic rats	150 mg/kg, 30 days	Decrease in blood glucose level more than glibenclamide	[340]
*Rheum ribes*	Root	Water, decoction	Alloxan diabetic rats	200 mg/kg, 30 days	Increased activity of *β*-cells	[344]

**Table 14 molecules-26-00742-t014:** Ethnobotany of plants from Jordan.

Scientific Name	Common Name	Family	Traditional Use	Part Used	Reference
*Achillea santolina*	Kaisoom, jeaidatelsabian	Asteraceae	Colic, cold, depurative, diabetes, kidney stones,	Aerial parts	[351,366]
*Artemisia herba alba*	Shaih	Asteraceae	Fever, menstrual and nervous problem	Aerial parts, root	[367]
*Artemisia sieberi*	Shaih	Asteraceae	Antispasmodic, antiarthritis, diabetes, pectoral	Foliage	[368]
*Arum dioscoridis*	Louf	Araceae	Anticancer	Leaf	[369]
*Arum palaestinum*	Louf	Araceae	Anticancer, internal bacterial infections, poisoning, disturbances of the circulatorysystem, cooking	Leaf	[369]
*Crataegus aronia*	Zaeroor	Rosaceae	Cardiovascular diseases, diuretic, hypertension, hyperlipidemia, kidney stone, laxative	Leaf	[370,371]
*Cichorium pumilum*	Hendba	Asteraceae	Antiseptic, antidiabetic, eczema	Flower, root	[372]
*Eryngium reticum*	Gersaana	Apiaceae	Scorpion and snake bite	Root	[371]
*Geranium graveolens*	Utryye	Geraniaceae	Diuretic, diabetes, stomachic	Leaf	[351]
*Pistacia atlantica*	Botom	Anacardiacae	Asthma, cough, diabetes, stomach ache,	Fruit, leaf, resin	[373,374]
*Rheum ribes*	Rabbas	Polygonaceae	Diabetes, hypertension, kidney sand and stones, obesity,	Root	[351]
*Teucrium polium*	Ja’deh	Lamiaceae	Abdominal pain, diabetes, urinary tract infection,	Aerial parts, shoot, leaves	[351]
*Varthemia iphionoides*	Qtteileh	Asteraceae	Abdominal pain, diabetes	Shoot, leaf	[351]

**Table 15 molecules-26-00742-t015:** Medicinal plants of Jordan constituents, use.

Scientific Name	Phytochemical Constituent	Pharmacological Use	Reference
*Achillea santolina*	Flavonoids, terpenoids, essential oils	Anti-inflammatory, immunomodulatory, antibacterial	[375]
*Artemisia herba alba*	Sesquiterpene lactones, phenolic compounds, essential oil, flavonoid	Antioxidant, antimicrobial, antivenom, antispasmodic, anthelmintic, diabetes, nematicidal, neurological activity	[376]
*Artemisia sieberi*	Essential oil, flavonoid, polyphenolic compounds	Antioxidant	[353]
*Arum dioscoridis*	Flavonoids, phenolic acids	Antioxidant, antimicrobial	[355,369]
*Arum palaestinum*	Pyrrole alkaloid	Antioxidant, anticancer, antidiabetic	[369,377]
*Cichorium pumilum*	Flavonoids, coumarins, caffeic acid derivatives, sesquiterpene lactones	Antitumor	[378]
*Eryngium reticum*	Coumarine, essential oils, terpenes, sesqui and monoterpenes, sitosterols, tannins, resins	Anti-snake and anti-scorpion venom, antibacterial, antifungal, antimalaria, antileshmania, antioxidant, antimutagenic, antihyperglycemic, cytotoxic	[359]
*Geranium graveolens*	Essential oil	Antiemetic activity, antioxidant, fumigant repellent	[374]
*Phaseolus vulgaris*	Alkaloids, anthocyanin, catechin flavonoids, saponins, tannins, terpenoids	Antioxidant, anticancer	[363]
*Pistacia atlantica*	Flavonoids, phenolic compounds, terpenoids, mono and sesquiterpenoids, volatile oil	Antimicrobial, antiviral, antidiabetic, antitumor, anticholinesterase activity, antioxidant	[374]
*Tecoma stans*	Alkaloid, chlorogenic acid	Diabetes, lower cholesterol,	[358,379]
*Teucrium polium*	Essential oil, flavonoid, Iridoids, steroidal compounds	Antioxidant, anti-inflammatory, anticancer antihyperlipidemic, antimicrobial, antimutagenic, antiulcer, antispasmodic, antinociceptive, hepatoprotective, diabetes	[258]
*Varthemia iphionoides*	Sesquiterpene, essential oil Flavonoids	Antiplatelet, antioxidant, antitumor, antibacterial, antifungal	[375]

**Table 16 molecules-26-00742-t016:** Plants with potential antidiabetic activity form Jordan.

Scientific Name	Part Used	Extraction Method, Solvent	Target	Intervention and Duration	Observations	Ref
*Achillea santolina*	Aerial parts	Water, reflux	Starch loaded rats	125, 500 mg/kg	Enhanced oral glucose tolerance	[343]
	Aerial parts	Water, reflux	MIN6 *β*-cell line	0.05–1 mg/mL	Dose dependent pancreatic *β*-cell proliferation	[350]
*Artemisia herba alba*	Aerial parts	Water, maceration	Alloxan diabetic rabbits	1 day	Significant hypoglycemic effect	[380]
*Artemisia sieberi*	Aerial parts	Essential oil, hydrodistillation	Alloxan diabetic rats	80 mg/kg, 30 days	LD_50_ 800 mg/kg (body weight)	[353]
*Arum dioscoridis & Arum palaestinum*	Aerial parts	Water, ethanol, reflux	Fasted rats	125, 250, 500 mg/kg	Concentration-dependent pancreatic lipase inhibition	[355]
*Crataegus aronia*	Flower, leaf	Water, reflux	Highcholesterol diet fed rats	100, 200, 400 mg/kg, 10 weeks	Potent antiobesity, marked triacylglycerol-reducing efficacy	[357]
*Cichorium pumilum*	Leaf	Ethanol, percolation	Alloxan diabetic rats	1 gm/kg	Reduced bloodglucose after 3 h of administration	[358]
*Eryngium creticum*	Aerial parts	Water, reflux	MIN6 *β*-cell line	0.1, 0.5, 1 mg/mL	Dose-dependent highly significant pancreatic *β*-cell proliferation	[350]
	Aerial parts	Water, reflux	Starch treated rats	125, 250, 500 mg/kg	Lack of effect on enzymatic starch digestion, no overall glycemicexcursion, no improvement in OGTT	[360]
*Geranium graveolens*	Leaf	Water reflux	Pancreatic *β*-cells MIN6	0.01–1.0 mg/mL	Augmented *β*-cell mass expansion, dose-dependent dual inhibition of *α*-amylase and *α*-glucosidase	[360,361]
*Phaseolus vulgaris*	Fresh pods	Ethanol, percolation	Alloxan diabetic rats	1 gm/kg	Significantly reduced blood glucose after 3 h of administration	[358]
*Pistacia atlantica*	Aerial parts	Water, reflux	Starch-loaded rat	125 mg/kg	Hypoglycaemic effect reversed at higher doses for the extract	[343]
	Aerial parts	Water, reflux	MIN6 *β*-cell line	0.01, 0.1, 0.5 mg/mL	Augmented acute *β*-cell insulin secretory efficacy, preserved *β*-cell integrity	[350]
*Rheum ribes*	Root, rhizome	Water, reflux	Pancreatic *β*-cells MIN6	0.01, 0.05, 0.1 and 0.5 mg/mL	Acute insulin secretion, augmented *β*-cell mass expansion	[260]
*Sarcopoterium spinosum*	Aerial parts	Water, reflux	Pancreatic *β*-cells MIN6	0.01–1.0 mg/mL	Insulinotropic and proliferative effects in the pancreas, dose-dependent dual inhibition of *α*-amylase and *α*-glucosidase	[343,361]
*Tecoma stans*	Leaf	Ethanol, percolation	Alloxan diabetic rats	1gm/kg	Significantly reduced bloodglucose after 3 h of administration	[358]
*Teucrium polium*	Aerial parts	Ethanol, percolation	Alloxan diabetic rats	1gm/kg	Reduced blood glucose after 3 h of administration	[358]
*Varthemia iphionoides*	Aerial parts	Water, reflux	Pancreatic *β*-cells MIN6	0.01–1.0 mg/mL	Augmented *β*-cell mass expansion	[361]

**Table 17 molecules-26-00742-t017:** Ethnobotany of Lebanon.

Scientific Name	Common Name	Family	Traditional Use	Part Used	Reference
*Centaurea horrida*	Not mentioned	Asteraceae	Diarrhea, diabetes, hypertension	Not mentioned	[387]
*Inula viscosa, Inula vulgaris*	Tayyoun, ‘Ergel-tayyoun	Asteraceae	Rheumatoisim	Whole plant	[388]
*Psoralea bituminosa*	Homan Homri	Fabaceae	Iintestinal ailments, gastric ulcers	Leaf, fruit	[389]
*Salvia libanotica*	Aiza’an Kassiin ‘Ouaiss’e Maryamiyy’e	Lamiaceae	Asthma, arthritis, antiseptic, aphrodisiac, antimicrobial, carminative, constipation carminative, cough, diabetes, expectorant, influenza, hypertension, gastralgia, hepatitis, febrifuge, nephropathy, rheumatism, spasmolytic, stomachic, stimulating memory	Flower, leaf	[390]

**Table 18 molecules-26-00742-t018:** Medicinal plants of Lebanon reported constituents, use.

Scientific Name	Phytochemical Constituent	Pharmacological Use	Reference
*Centaurea horrida*	Flavonoids, lactones, phenolic acids	Antioxidant	[387]
*Inula viscosa, Inula vulgaris*	Costic acid, flavonoid, guaianolide, phenolic compounds, terpenoid	Abortifacient, antibacetraial, anti-implantation, antihypertensive, cytotoxic, hypoglycemic	[383]
*Psoralea bituminosa*	Furanocoumarins, isoflavonoid, meroterpenoids, sesquiterpene, volatile oil	Antioxidant, cytotoxicity	[386]
*Saliva libanotica*	Essential oil	Antioxidant activity, anticholinesterase activity, anticancer	[391]

**Table 19 molecules-26-00742-t019:** Plants with antidiabetic potential from Lebanon.

Scientific Name	Part Used	Extraction Method, Solvent	Target	Intervention and Duration	Observations	Ref
*Centaurea horrida*	Root	Ethanol 80%, maceration	Alloxan diabetic rats	25, 50, 100 mg/kg, 8 days	Improve peripheral nerve function	[381]
*Hordeum spontaneum*	Root	Ethanol 80%, maceration	Alloxan diabetic rats	25, 50, 100 mg/kg, i.p, 8 days	Improve peripheral nerve function	[381]
*Inula viscosa, Inula vulgaris*	Aerial parts	Ethanol, maceration	Alloxan diabetic rats	12.5, 25 and 50 mg/kg, 8 days	Long treatment led to reverse free radicals activities	[384]
*Psoralea bituminosa*	Aerial parts	Ethanol (80%), ethyl acetate, hexane, maceration	Alloxan diabetic rats	25, 50, 100 mg/kg, 8 and 21 days	Significant management of neuropathic pain	[386]
*Rheum ribes*	Rhizome	Water, maceration	Alloxan diabetic rats	12.5, 25, 50 mg/kg, 8 days	Had a protective effect against diabetes and diabetic neuropathy	[381]
*Salvia libanotica*	Root	Ethanol 80%, maceration	Alloxan diabetic rats	12.5, 25, 50 mg/kg, 8 days	Improve peripheral nerve function	[381]

**Table 20 molecules-26-00742-t020:** Medicinal plants of Palestine ethnobotany.

Scientific Name	Common Name	Family	Traditional Use	Part Used	Reference
*Atriplex halimus*	Katf	Chenopodiaceae	Diabetes, heart disease	Leaf	[406]
*Ocimum basilicum*	Rehan	Lamiaceae	Diarrhea, kidney stone, carminative, diuretic, dysentery, skin disease, anti-colic.	Whole palnt	[407,408]
*Sarcopoterium spinosum*	Bullan, Natesh	Rosaceae	Anti-inflammatory, toothache, analgesic, for hemorrhoids, antidiabetic, stomach pain, diuretic, renal calculi, stimulant for circulation	Root, fruit, seed	[409]
*Trigonella foenum-graecum*	Hilbeh	Fabaceae	Diabtes, cancer	Seed	[410,411]
*Withania somnifera*	Samoh	Solanaceae	Skin disease, kidney stone, wound healing	Leaves, root, young branches	[412,413]

**Table 21 molecules-26-00742-t021:** Medicinal plants of Palestine reported constituents, use.

Scientific Name	Phytochemical Constituent	Pharmacological Use	Reference
*Atriplex halimus*	Alkaloid, flavonoid, saponin, sterol, tannin	Antidiabetic, antimicrobial, antioxidant, insecticidal	[392]
*Ocimum basilicum*	Cardiac glycosides, caffeic acid derivatives, essential oil, flavonoid, phenolic acids, saponins, tannins	Antimicrobial, hepatoprotective, hypertension nematicidal	[393,394]
*Sarcopoterium spinosum*	Catechin, epicatechin, essential oil	Antidiabetic	[414]
*Trigonella foenum-graecum*	Alkaloid, flavonoid, polyphenols, saponin, volatile oil	Antidiabetic, antilipidemic, anticarcinogenic, anticataract, antioxidant, immunomodulatory	[399,400,415]
*Withania somnifera*	steroidal alkaloids and lactones, withanolides	Anticancer, immunomodulatory, anti-inflammatory, antistress, adaptogenic, CNS activities, cardiovascular activities	[416]

**Table 22 molecules-26-00742-t022:** Plants with antidiabetic potential from Palestine.

Scientific Name	Part Used	Extraction Method, Solvent	Target	Intervention and Duration	Observations	Ref
*Atriplex halimus*	Leaf, stem	Ethanol (50%) extract, maceration	HepG2, L6myc cell line	0–2 mg/mL	Increased GLUT4 translocation, EC50 was about 2 mg/mL	[272]
*Gundelia tournefortii*	Aerial parts	Methanol, hexane extract, maceration	Rat L6 muscle cell lines	0–1 mg/mL	Methanol extract was the most efficient in GLUT4 translocation enhancement in skeletal muscle	[417]
*Ocimum basilicum*	Aerial part	Methanol, hexane, dichloromethane extracts, reflux	L6 muscle cell line	0–2 mg/mL	The extracts increased GLUT4 translocation up to 7 times	[394]
*Olea europaea*	Leaf	Ethanol(80%) extract, soxhlet	Streptozocin diabetic rats	10, 20, 40 mg	The extract inhibited both digestion and absorption (concentration-dependent)	[418]
*Sarcopoterium spinosum*	Root	Aqueous extract, decoction	RIN *β*-cells, L6 myotubes, 3T3-L1 adipocytes, AML-12 hepatocytes	0.001–10 mg/mL	Preventive effect for progression in diabetes	[395]
	Root	Aqueous extract, decoction	KK-a/y mice	600 mg/kg, 6 weeks	Improved insulin sensitivity, increased glucose uptake	[396]
	Root, leaf, fruit	Aqueous extract, decoction	3 T3-L1 adipocytes cell line	Root: 1 mg/mLFruit: 2 mg/mL	Inhibited *α*-glucosidase and amylase, induced insulin secretion	[397]
	Root	Aqueous extract, decoction	High-fat diet mice, KK-Ay male mice	70 mg/day, 6 weeks	Improved glucose tolerance and sensitivity	[398]
*Trigonella foenum-graecum*	Seed	Ethanol (50%) extract, maceration	HepG2, L6myc cell line	0–2 mg/mL	Significant translocation of GLUT4	[272]
*Urtica dioica*	Leaf, stem	Ethanol (50%) extract, maceration	HepG2, L6myc cell line	0–2 mg/mL	Extract almost doubled GLUT4 translocation	[272]
*Withania somnifera*	Leaf, root	Methanol extract, maceration, isolated compound	Pancreatic RIN-5F	50 mg/mL	Extracts increased glucose uptake in myotubes and adipocytes (dose-dependent). Leaf extract increased insulin secretion, Withaferin A increased glucose uptake.	[403]

**Table 23 molecules-26-00742-t023:** Ethnobotany of plants from Turkey.

Scientific name	Common Name	Family	Traditional Use	Part Used	Reference
*Cichorium intybus*	Sakızotu, hindiba, sakızçiçeği	Asteraceae	Cancer, kidney stone, hemorrhoids, urinary disorders, wound keeling	Aerial parts, leaf, root	[202,428]
*Cinnamomum verum*	Tarçın	Lauraceae	Not mentioned	Not mentioned	[429]
*Haplophyllummyrtifolium*	Not mentioned	Rutaceae	Sudanese and Mongolian folk medicin: antipyretic, diarrhea	Not mentioned	[430]
*Helichrysum graveolens*	Olmezcicek or altinotu	Asteraceae	diuretics, as lithagogues, for stomachache, for anti-asthmatic properties, against kidney stones	Not mentioned	[431,432]
*Hedysarum varium*		Fabaceae			[420]
*Laurus nobilis*	Define	Lauraceae	Antiseptic, against rheumatic pain	Fruit, seed	[433,434]
*Mentha pulegium*	Filiskin, yarpuz	Lamiaceae	Bronchitis, cold, flatulent dyspepsia, intestinal colic, chlorera, food posioning, sinusitis, tuberculosis	Flowering aerial parts	[425,435,436]
*Phlomis armeniaca*	Calba	Lamiaceae	Colds, stomachache	Not mentioned	[419]
*plantago lanceolata*	Giyamambel	Plantaginaceae	Diabetes, stomach ache	Not mentioned	[419]
*Onobrychis hypargyrea*	Not mentioned	Fabaceae	Cold and flu	Not mentioned	[420]
*Origanum onites*	Kekik’	Lamiaceae	Abdominal ache, bronchitis, cold, diabetes, dizziness, headache, hypertension, high cholesterol, itching, gastralgia, leukemia, toothache, stomach disorders	Aerial parts flowers, leaves	[424]
*Salvia limbata*	Baresaspi	Lamiaceae	Colds, stomach ache, wounds	Not mentioned	[419]

**Table 24 molecules-26-00742-t024:** Ethnobotany of plants from Turkey.

Scientific Name	Common Name	Family	Traditional Use	Part Used	Reference
*Cistus laurifolius*	Defne yapraklı	Cistaceae	Diabetes, fever in common cold, rheumatic and inflammatory diseases, peptic ulcer, applied externally in a line of the kidneys for urinary inflammations	Leaf, flower, bud, branch	[437,446]
*Juniperus oxycedrus*	KatranArdıcı	Cupressaceae	Abdominal pain bronchitis, calcinosis, common cold, cough, diabetes, gynecological diseases, fungal infections, hemorrhoids, kidney stone, stomachic disorders, kidney inflammation, wounds	Tar, leaf, fruit, berry	[438,447]
*Origanum minutiflorum*	Toga kekik, mountain kekik,	Lamiaceae	Herbal tea	Leaf	[444,448]
*Rhus coriaria*	Sumac	Anacardiacea	Diabetes, toothache, eye diseses	Not mentioned	[449,450]
*Salvia triloba*	AnadoluAdacay	Lamiaceae	Blood, skin andinfectious, ailments of the digestive, circulatory and respiratory systems	Leaf	[445]
*Thymus praecox*	Kekik	Lamiaceae	Herbal tea, condiment	Herbal parts	[451,452]

**Table 25 molecules-26-00742-t025:** Medicinal plants of Turkey reported constituents, use.

Scientific Name	Phytochemical Constituent	Pharmacological Use	Reference
*Cistus laurifolius*	Flavonoids, phenolic acids	Antiulcerogenic, analgesic, antioxidant, hepatoprotective	[437]
*Juniperus oxycedrus*	Lignans, abietane, sesquiterpenes, diterpenes, biflavonols, tannins, coumarins, flavonoids, sterols, terpenes, mono, sesqui and diterpenes, alkanes, fatty acids, waxes	Antioxidant, antiseptic, antiviral, anti-inflammatory, analgesic, anticancer, antidiabetic, neuroprotective	[438,447,453]
*Origanum minutiflorum*	Essential oils	Antioxidant, antibacterial, antiviral, antifungal	[444]
*Rhus coriaria*	Anthocyanins, flavonoids, isoflavonoids, tannins, terpenoids	Antioxidant, antiseptic, antifungal, antibacterial, anti-ischemic, antifibrogenic, antitumourigenic activities, hypoglycemic, non mutagenic, fever, DNA protective, hypouricaemic, hepatoprotective properties	[449,454]
*Salvia triloba*	Essential oil, flavonoids, flavone, terpene, sesqui, di and triterpenes, steroidal compounds	Antimicrobial	[452,455]
*Thymus praecox*	Essential oil	Anti-inflammatory, antifungal, antiviral, antioxidant, anticancer, antidiabetic effects	[445,452]

**Table 26 molecules-26-00742-t026:** Plants with antidiabetic potential from Turkey.

Scientific Name	Part Used	Extraction Method, Solvent	Target	Intervention and Duration	Observations	Ref
*Cistus laurifolius*	Leaf	Water, ethanol, maceration	Streptozocin diabetic rats	250, 500 mg/kg	Inhibited of *α*-glucosidaseand *α*-amylase	[437]
*Heracleum persicum*	Whole plant	Water, maceration	Streptozocin diabetic rats	100, 200, 400 mg/kg 21 days	Decreased HbA1c, restored diabetic complications parameters towards normal, increased AST and ALT significantly	[320]
*Juniperus oxycedrus*	Berries	Water, maceration isolated compound	Streptozocin diabetic rats	500, 1000 mg/kg Shikimic acid: 15 and 30 mg/kg, 8 days	The effect of shikimic acid was more effective than the reference antidiabetic drug glipizide, TG and enzyme levels were reduced significantly	[438]
	Leaf	Water, ethanol, maceration	Streptozocin diabetic rats	500, 1000 mg/kg	Fatty acids were found as the major compounds	[440]
	Fruit, leaf	Ethanol 80%, maceration	Streptozocin diabetic rats	500, 1000 mg/kg, 10 days	Augment Zn level in liver	[441]
*Juniperus foetidissima* & *Juniperus sabina*	Leaf, fruit	Ethanol 80%, maceration	Streptozocin diabetic rats	500, 1000 mg/kg, 8 days	Leaf extract caused death at both doses	[439]
*Origanum minutiflorum*	Whole plant	Water, decoction	Streptozocin diabetic rats	150 mg/kg, 30 days	Decreased ALT and AST levels	[444]
*Salvia triloba*	Aerial parts	Methanol 80%, soxhlet	Streptozocin-nicotinamide diabetic rats	100, 200 mg/kg, 21 days	Little weight loss	[445]
*Thymus praecox*	Aerial parts	Methanol, maceration	Streptozocin-nicotinamide diabetic rats	100, 200 mg/kg, 21 days	Little weight loss	[445]

## Data Availability

Data available in a publicly accessible repository.

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
