# Peer review of "Middle East Medicinal Plants in the Treatment of Diabetes: A Review"

_molecules, 2021, doi:10.3390/molecules26030742_

Round 1
Reviewer 1 Report
Alaa M. Abu-Odeh and Wamidh H. Talib's “Middle East medicinal Plants in the treatment of diabetes: a review” provides a wealth of evidence on medicinal plants used to treat diabetes. Undoubtedly, this review will be of interest to both medicinal chemists and biologists developing antidiabetic drugs.
However, a number of comments of varying degrees of significance can be made.
The review contains a large number of typos and factual errors. First of all, words are often written together, for example, "inhibitoryactivity", "supplementationof" and many others. There are errors in the names of the compounds (for example, acrabose - need acarbose). A number of statements require clarification or correction, such as “In particular, naringenin (naringin counting for more than 50% in Ajuga)”, p. 2. Is naringin more than 50% dry weight or flavonoid content? What do the authors describe naringenin or naringin?
In some cases, citation of the literature does not follow the rules - for example, “Two species of this genus are endemic in South America and one species in tropical Africa (Fan and Xiang, 2001; Xiang et al., 2006), p. 29”.
Figure 1. Botanical families studied - inconsistency between text and diagram - in text Fabaceae 21 species, in the diagram Fabaceae 9 species.
It seems to me that it would be useful to give the names of compounds present in plants and responsible for their activity, and not be limited to mentioning classes, such as “alkaloid, anthocyanin, catechin, ellagitannins, flavonoid, sterol, tannins” (p. 42).
The most important remark relates to the construction of the review. What is the basis for the division of plants by country, if the distribution area of most plants listed in the review are "all Mediterranean countries, South Western Asia, Europe and North Africa, p. 53"? In the text of the review, there is no indication of the specific use of any plants in a certain country. Would it be more convenient to group the data by plant family?
Author Response
Thank you for your constructive comments. All comments were considered and changes in the manuscript were done. Please see attached

Reviewer 2 Report
The review describing the antidiabetic role of the various plant species present in the Middle East region. A thorough revision is required before accepting the MS for publication. The author must address the following:
- A review paper is too lengthy. I suggest to reduce the text and place important information’s in the form of tables and figures.
- A thorough revision of the English language and grammar needs to address.
- The abstract must be rewritten with some concrete information
- The name of the plant must be written scientifically throughout the manuscript
- Address typographical mistakes eg. pp2 "In the present.... Middle East area.the ...........................future research. pp4 (subheading 2.3.1) 2 mg/Gm
- Write the full name of abbreviations while appearing first in the review. e.g pp2 KSA????
- Remove the repetition of Bodyweight that appears in several places of the manuscript such as pp2 (subheading 2.1.1); pp 3 (subheading 2.1.2) and pp 4 (subheading 2.3.1).
- Properly mention Y-axis in Figures 3 and 4.
- Make figure 5 again in a more informative way.
- It would be better if the author provides some flow diagram/pathway demonstrating the mechanistic role of plants as a curative agent of diabetes.
Author Response

(The authors gave the same response as above.)

Round 2
Reviewer 2 Report
The required changes has been done. I recommend to publish the article.